# Spatial signatures for predicting immunotherapy outcomes using multi-omics in non-small cell lung cancer

Thazin N. Aung [1,10], James Monkman [2,10], Jonathan Warrell [3,4,10], Ioannis Vathiotis[1], Katherine M. Bates [1], Niki Gavrielatou[1], Ioannis P. Trontzas[1], Chin Wee Tan [2,5,6], Aileen I. Fernandez[1], Myrto Moutafi[1], Ken O' Byrne[7], Kurt A. Schalper [1], Konstantinos Syrigos[8], Roy S. Herbst [9], Arutha Kulasinghe [2] ✉ & David L. Rimm [1] ✉

Non-small cell lung cancer (NSCLC) shows variable responses to immunotherapy, highlighting the need for biomarkers to guide patient selection. We applied a spatial multi-omics approach to 234 advanced NSCLC patients treated with programmed death 1-based immunotherapy across three cohorts to identify biomarkers associated with outcome. Spatial proteomics ($n = 67$) and spatial compartment-based transcriptomics ($n = 131$) enabled profiling of the tumor immune microenvironment (TIME). Using spatial proteomics, we identified a resistance cell-type signature including proliferating tumor cells, granulocytes, vessels (hazard ratio (HR) = 3.8, $P = 0.004$) and a response signature, including M1/M2 macrophages and CD4 T cells (HR = 0.4, $P = 0.019$). We then generated a cell-to-gene resistance signature using spatial transcriptomics, which was predictive of poor outcomes (HR = 5.3, 2.2, 1.7 across Yale, University of Queensland and University of Athens cohorts), while a cell-to-gene response signature predicted favorable outcomes (HR = 0.22, 0.38 and 0.56, respectively). This framework enables robust TIME modeling and identifies biomarkers to support precision immunotherapy in NSCLC.

Patients with advanced non-small cell lung cancer (NSCLC) without targetable driver alterations generally face a poor prognosis, with only a minority responding to existing therapies. More than 50% develop resistance, and approximately 40% show primary resistance even with high programmed death 1 (PD-1) ligand 1 (PD-L1) expression[1–3]. Furthermore, immunotherapy can cause severe immune-related toxicities, including fatal events. These challenges highlight the urgent need for patient stratification to optimize therapeutic efficacy.

Spatial multi-omics enhances our understanding of the tumor immune microenvironment (TIME) by integrating various omics data in the context of the underlying tissue architecture. This approach combines protein cell phenotyping with gene expression profiling,

delivering unique insights into cellular functions and disease mechanisms. Techniques such as codetection by indexing (CODEX)[4] enable high-resolution protein mapping in intact tissues, revealing both cellular heterogeneity and spatial dynamics within the TIME. Similarly, Digital Spatial Profiling (DSP)-GeoMx Whole Transcriptome Analysis (WTA) enables spatial transcriptomic profiling at cellular compartment resolution, allowing direct correlation between gene expression and spatially localized cellular phenotypes[5]. Together, spatial proteomics and transcriptomics offer a comprehensive view of tissue organization, uncovering key molecular interactions and disease-associated changes[6]. This integrated strategy enables the identification of compartment-specific biomarkers and therapeutic

targets for diagnostic and prognostic use[7]. Furthermore, machine learning approaches have shown that complex spatial molecular patterns can be linked to clinical outcomes[8]. Predicting the outcomes of first-line immunotherapy treatments is particularly important due to the absence of confounding from prior treatments. Tailoring models to predict treatment outcomes specifically for first-line immunotherapy aids in clinical decision-making by ensuring that assessments are unaffected by prior treatments, thereby enhancing treatment efficacy and enabling predictions for patients without multiple treatment exposures.

Our study aimed to develop a robust machine learning approach for training spatial signatures to predict first-line immunotherapy outcomes, including progression-free survival (PFS) at 2 and 5 years, and overall survival (OS), in NSCLC using a multi-omics approach. We hypothesized that integrating cell types and gene expression data with spatial context would enhance predictive accuracy. We first constructed cell-type-based signatures, then derived gene signatures from outcome-associated cell types using a training cohort (Yale cohort; Fig. 1a). These resistance and response signatures were validated in two independent external cohorts from University of Queensland and University of Athens (UQ and Greek cohorts respectively; Fig. 1a). By integrating spatial omics and machine learning techniques (Fig. 1b), we aim to improve treatment decisions and patient outcomes.

## Results

### Cell fraction association with PFS

Spatial proteomic profiling of advanced NSCLC tissues identified distinct cellular populations within tumor and stromal compartments. Figure 2a and Extended Data Fig. 1a,b illustrate cell phenotyping using a 29-marker panel, while Fig. 2b lists the markers used for identifying specific cell types. Representative tumor areas of interest (AOIs) are displayed in Fig. 2c,d. Tumor cells were most abundant in tumor AOIs, whereas M1/M2 macrophages and fibroblasts predominated in stromal AOIs. We then compared cell-type distributions as a function of PFS. In tumor AOIs from the training (Fig. 2e) and validation cohorts (Fig. 2f), granulocytes and proliferating tumor cells were enriched in patients with shorter PFS. In stromal AOIs (Fig. 2g,h), M1 and M2 macrophages were more prevalent in patients with longer PFS. These trends were consistent across cohorts. We also analyzed cell composition in the Yale cohort using a 5-year PFS cut point. Due to limited follow-up, this analysis was not performed in the UQ cohort. The observed patterns aligned with earlier findings that granulocytes and proliferating tumor cells were enriched in progressors in the tumor compartment, while M1 and M2 macrophages were more prevalent in nonprogressors in the stromal compartment (Supplementary Fig. 1a,b).

### Univariable association of cell fraction with 2-year PFS

To further explore associations between cell fractions and treatment outcome, we used a 2-year PFS endpoint and performed univariable Cox analyses for each cell type and compartment independently. In the tumor compartment, granulocytes, proliferating tumor and vessel cells were linked to increased risk of disease progression or death (Fig. 3a and Supplementary Fig. 2a). These trends were observed across cohorts but did not reach statistical significance after Benjamini–Hochberg (BH) adjustment. Figure 3b shows a representative region of interest (ROI) from a patient with shorter PFS, enriched in granulocytes and vessel cells. In the stromal compartment, M1 macrophages, M2 macrophages and CD4 T cells were associated with improved PFS in both cohorts (Fig. 3c and Supplementary Fig. 2b). While consistent trends were observed, for instance, granulocytes, vessels and proliferating tumor cells were linked to worse PFS, and M1 and M2 macrophages were associated with better PFS, none remained significant post-BH-adjustment. We also investigated associations with 5-year PFS in the Yale cohort (Supplementary Fig. 2c,d), which showed consistent patterns with the 2-year analysis across both tumor and stromal compartments.

### Spatial cell-type signatures for resistance

Spatial cell-type signatures were developed using spatial proteomic-derived cell fractions to predict outcomes to immunotherapy in advanced NSCLC patients. A schematic overview of the signature generation pipeline is shown in Extended Data Fig. 2, which is based on a previously established robust signature training approach[5]. The Yale cohort served as the training set, which was split multiple times into tenfolds. Least absolute shrinkage and selection operator (LASSO)-penalized Cox models were built to predict 2- and 5-year PFS, constrained to identify resistance-associated cell types by enforcing their coefficients to be non-negative (Methods). Each split generated a LASSO model using cross-validation, and a final Cox regression model was trained using cell types consistently selected across all splits, which identified proliferating tumor cells, vessels and granulocytes as high-risk features. The final model was evaluated on the full training cohort and then tested in the independent UQ validation cohort. In the tumor compartment of the training set, the resistance model was significantly associated with worse PFS (HR = 3.8, P = 0.004, two-sided log-rank test; Fig. 3d,e). Validation in the UQ cohort showed consistent predictive value (HR = 1.8, P = 0.05, one-sided log-rank test; Fig. 3f). In the stromal compartment, granulocytes and proliferating tumor cells were predictive in the training set (Supplementary Fig. 3a,b) but did not reach significance in the UQ cohort validation cohort (Supplementary Fig. 3c), despite similar trends. These results highlight the role of granulocytes, vessels and proliferating tumor cells in resistance within the tumor compartment, underscoring the spatial specificity of these signatures.

### Spatial cell-type signatures for response

Spatial cell-type signatures were developed using spatial proteomics-derived cell fractions to predict response to immunotherapy in advanced NSCLC patients. As with the resistance model, the Yale cohort was used for training, using a robust signature training approach[5]. Cell-type signatures were trained to predict 2- and 5-year PFS, constrained to identify response-associated cell types, by enforcing their coefficients to be nonpositive (Methods). In the stromal compartment, M1 and M2 macrophages and CD4 T cells had negative coefficients, indicating reduced risk. These were significantly associated with improved outcomes in the training cohort for PFS-5 years (HR = 0.4, P = 0.019, two-sided log-rank test) and showed a similar trend for PFS-2 years (HR = 0.57, P = 0.12; Fig. 3g,h). In the UQ validation cohort, this signature remained predictive (HR = 0.49, P = 0.036, one-sided log-rank test; Fig. 3i), supporting the role of stromal M1 and M2 macrophages and CD4 T cells mediating immunotherapy response. In contrast, the tumor compartment response model, which included tumor cells, M1 macrophages, M2 macrophages and DCs, was predictive in the training cohort (Supplementary Fig. 3d,e), but did not achieve significance in the validation cohort (Supplementary Fig. 3f). These results emphasize the importance of the stromal immune context in predicting immunotherapy benefit.

### Spatial cell-type signature association with OS

We further evaluated the association of the resistance model with OS at 2 and 5 years (Supplementary Fig. 4a–d) and observed consistent trends with PFS across cohorts. Conversely, when we examined the association of the response model with OS at 5 years and 2 years, the response model showed no statistically significant association in OS at either time point in the UQ validation cohort (Supplementary Fig. 5a–d). This suggests that response-associated cell types are more reflective of early treatment dynamics and disease control rather than long-term survival. The lack of statistical significance in OS highlights the temporal specificity of the response signature.

### Immune cell interactions and spatial niches in the TIME

To explore spatial features linked to response and resistance models derived from cell frequencies, we analyzed spatial organization

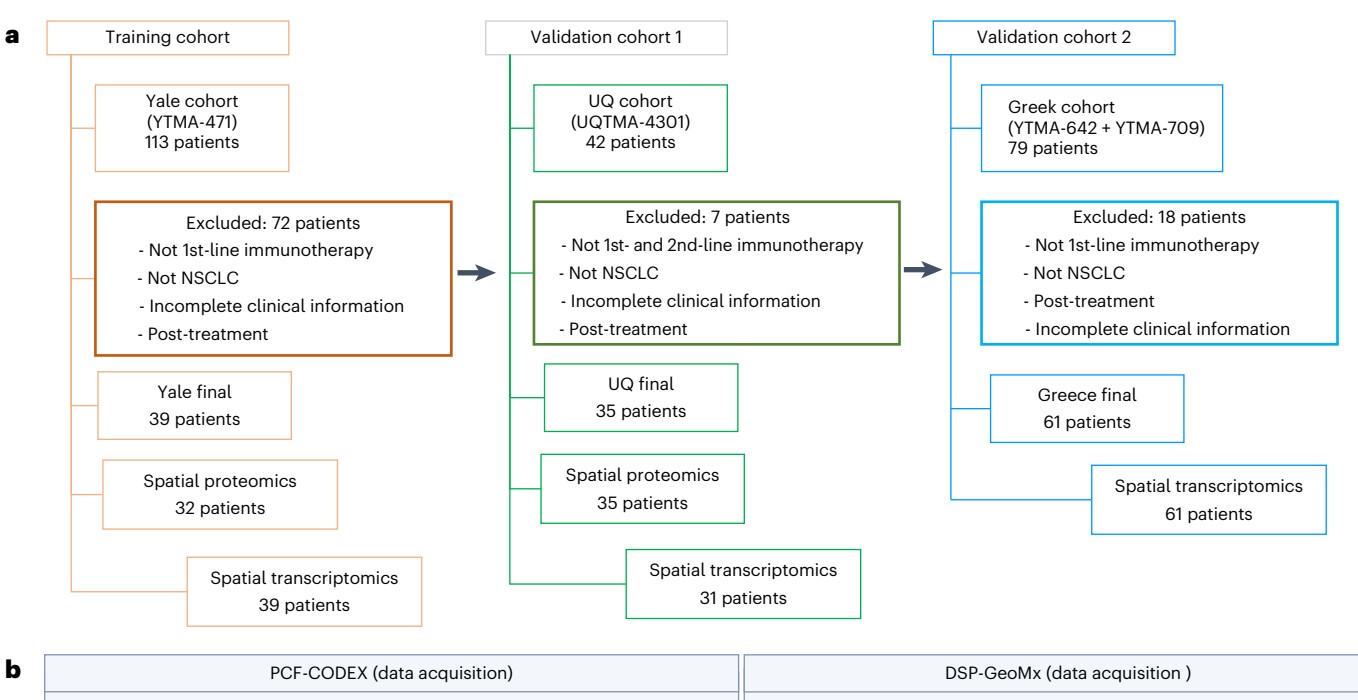

**Fig. 1 | Overview of study. a**, Flowchart of inclusion criteria for the Yale, UQ and Greek NSCLC cohorts is shown. This consort diagram presents the step-by-step procedure for the inclusion of participants. It outlines the criteria and decision-making steps that determine whether a participant qualifies for inclusion in the research. **b**, The schematic depicts the comprehensive workflow for the study. from the initiation through to the analysis and reporting of results. Panel **b** was created with Biorender.com.

and interactions. Representative Voronoi diagrams illustrate the spatial distribution of cell types associated with the response (Fig. 4a) and resistance (Fig. 4b) models. M2 macrophages predominated in response-associated regions, while the resistance Voronoi diagram shows an enrichment of vessels, PD-L1⁺ tumor cells and proliferating tumor cells. Cellular neighborhoods were constructed to determine

the spatial aggregation of cellular communities[9]. We identified ten distinct neighborhoods representing unique spatial architectures (Fig. 4c). These were annotated based on predominant cell-type enrichment, including M1 macrophage, vessel and immune cell, tumor cell, PD-L1⁺ tumor, granulocyte, tumor-adjacent stroma, immune-rich, M2 macrophage, proliferating tumor and stromal cell communities.

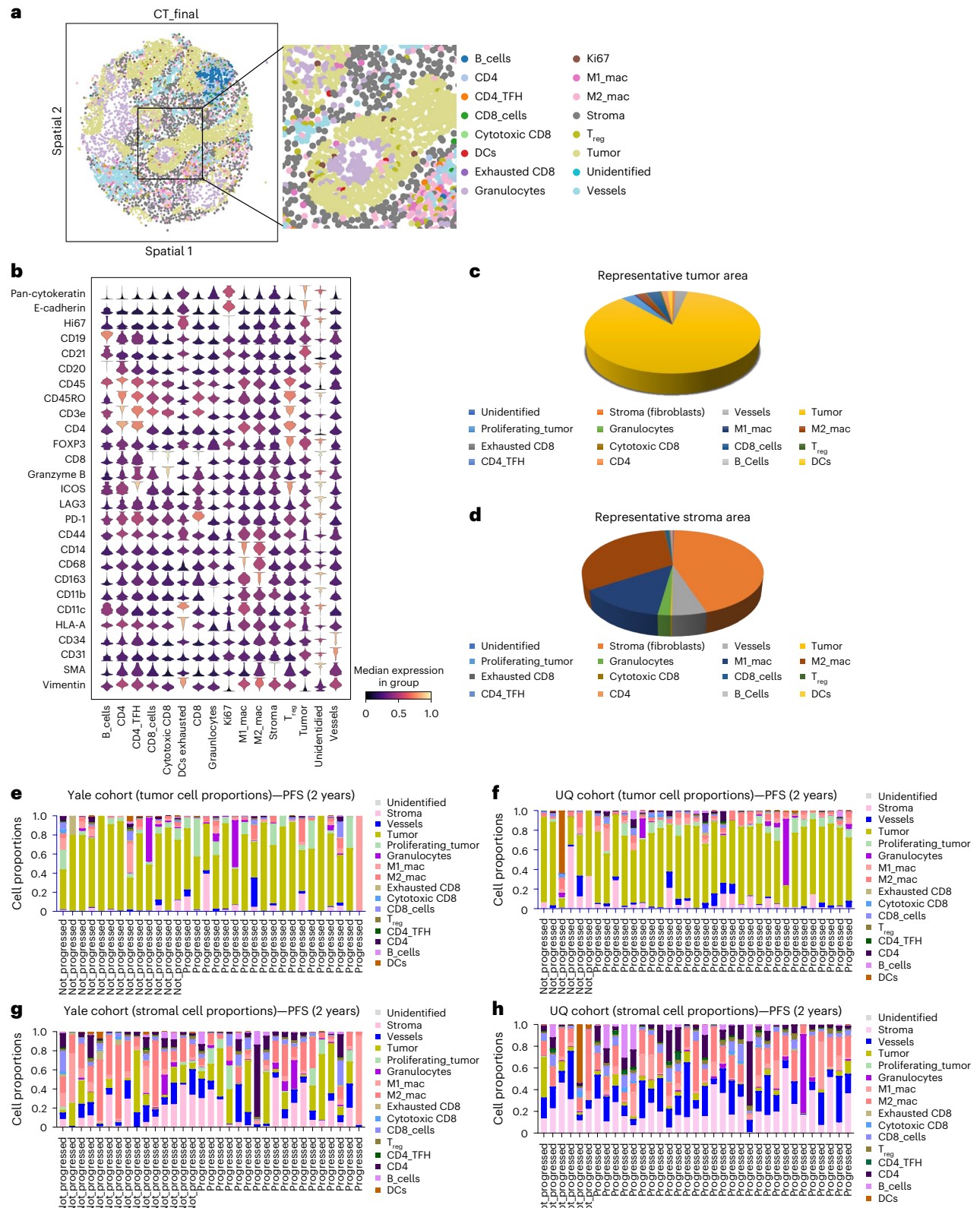

**Fig. 2 | Cell-type and composition of tumor and stroma compartments.**
**a**, A representative figure illustrates the different cell phenotypes in an ROI from a patient. **b**, The specific markers used in CODEX for identifying various cell types within the samples are listed. **c**, A representative tumor compartment from **a** highlights the cellular composition and diversity of cell types within the tumor. **d**, The variety of cell types present in the stromal compartment is shown. **e**, The proportions of different cell types within the tumor compartment of patients from the Yale cohort are grouped based on the 2-year PFS index, illustrating the

potential relationship between cellular composition and patient prognosis. **f**, A similar analysis is presented for the tumor compartment of the UQ cohort, grouped by the 2-year PFS index. **g**, The distribution of different cell types within the stroma compartment of patients from the Yale cohort is grouped by the 2-year PFS index. **h**, Similarly, the proportions of different cell types within the stroma compartment of the UQ cohort are grouped by the 2-year PFS index, highlighting the diversity and potential prognostic implications of stromal cell composition.

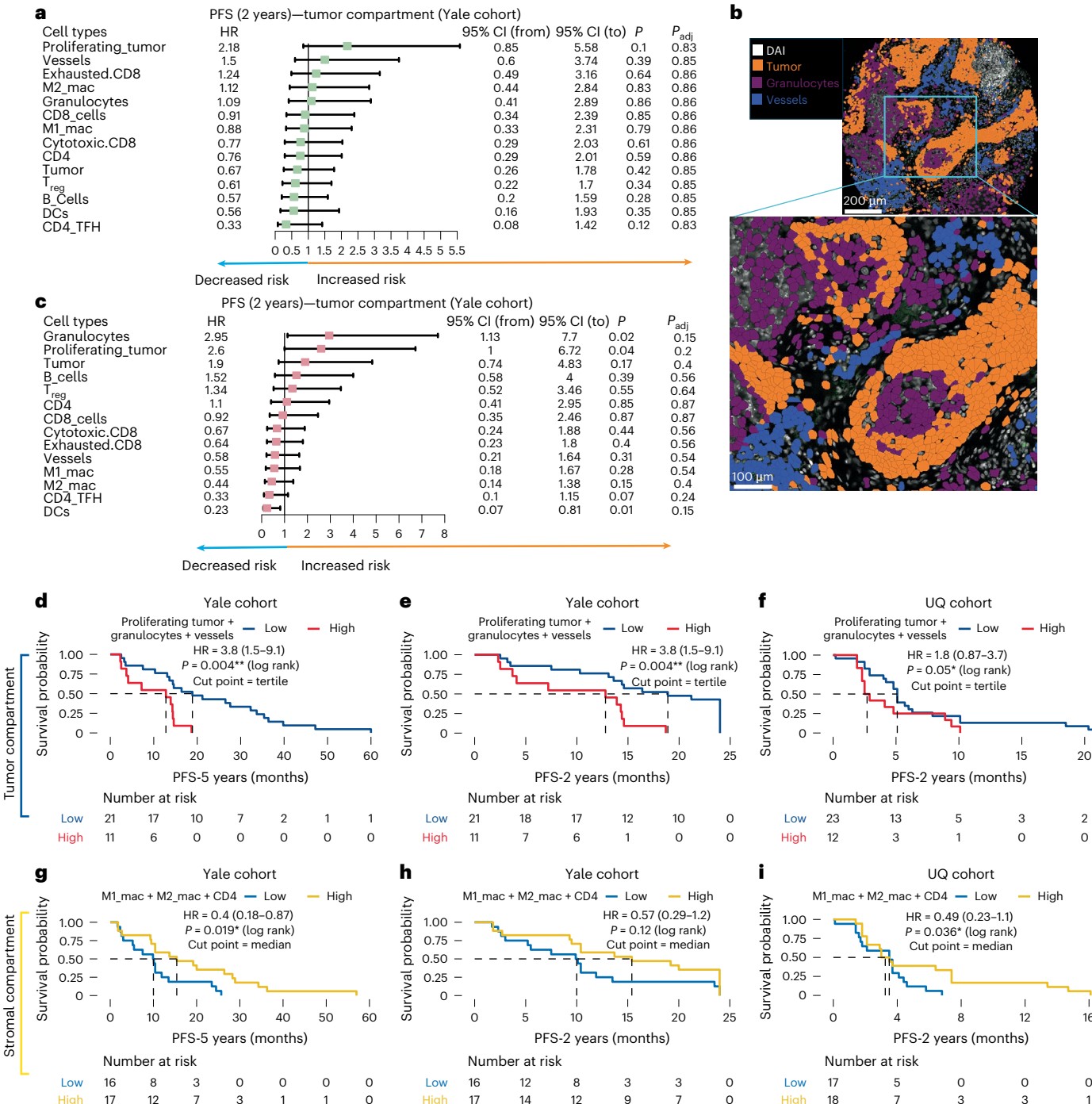

**Fig. 3 | Spatial cell-type signatures using PCF proteomics for resistance and response. a**, Univariable analysis of PFS at 2 years in relation to different cell types within the tumor compartment of the Yale cohort, using the median as a cut point. HRs and 95% CIs were derived from Cox proportional hazards models, and two-tailed log-rank $P$ values were calculated and BH adjustment applied. Each data point represents one ROI from each patient. A total of $n = 32$ patients were analyzed using spatial proteomics. **b**, A representative ROI from a patient who progressed, characterized by a high presence of granulocytes and vascular structures. **c**, Univariable analysis of PFS at 2 years in the stromal compartment of the Yale cohort, highlighting the impact of different cell types on disease progression over the 2-year timeframe, with the median used as a cut point. **d**, The Kaplan–Meier plot shows the performance of the resistance signature predicting

PFS at 5 years from the tumor compartment of the training cohort. **e**, Performance of the resistance signature predicting PFS at 2 years from the tumor compartment of the training cohort. **f**, Validation of the resistance signature predicting PFS at 2 years in the tumor compartment of the UQ validation cohort. **g**, Performance of the cell-type-response signature predicting PFS at 5 years from the stromal compartment of the training cohort. **h**, Performance of the response signature predicting PFS at 2 years from the stromal compartment of the training cohort. **i**, Validation of the response signature predicting PFS at 2 years in the stromal compartment of the UQ validation cohort. Two-tailed and one-tailed log-rank tests are used on the discovery and validation cohorts, respectively, where the direction of the effect in the latter is chosen to match the direction of the effect in the former. CI, confidence interval.

Neighborhood frequencies were correlated with PFS. Proliferating tumor communities were more common in patients who progressed ($P = 0.07$, two-tailed $t$ test), supporting the role of tumor proliferation in therapy resistance (Fig. 4d). Changes in neighborhood composition further revealed insights into the biology of therapy response. Within granulocyte-rich neighborhoods, higher levels of PD-L1+ tumor cells, B cells, and DCs were linked to progression ($P < 0.0001$, Ordinary Least Squares; OLS model $F$ test), suggesting a protumor microenvironment (Extended Data Fig. 3). Pairwise spatial interactions were analyzed to uncover cell–cell relationships (Fig. 4e). Proliferating tumor cells predominantly interacted with granulocytes and endothelial cells (vessels), implying roles in tumor proliferation and angiogenesis. In the response model, we examined interactions between M1/M2 macrophages, or CD4 T cells and other immune or tumor cells. B cells showed high interactions with M1 macrophages, suggesting a potential role in antitumor activity. M2 macrophages showed near-significant interaction with CD8 T cells ($P = 0.06$), highlighting a modulation of cytotoxic responses. Comparing interaction scores between progressors and nonprogressors revealed higher M2–CD4 T cell interactions and M2–CD8 T cell interactions in nonprogressors, (Fig. 4f), indicating an immune-supportive microenvironment in patients with better outcomes.

## Macrophage PD-L1 expression linked to immunotherapy response

The response model, derived from spatial analysis, highlights the critical role of M1 and M2 macrophages and CD4 T cells in predicting response to PD-1-based immunotherapy. Uniform Manifold Approximation and Projection (UMAP) visualizations of all 14 cell types (Fig. 4g), alongside expression dynamics of each marker, identified in spatial proteomic data, show higher PD-L1 expression on M1 (CD68+) and M2 (CD163+) macrophages compared to PanCK+Ki67+ proliferating tumor cells. In the Yale cohort, nonprogressors showed significantly higher PD-L1 expression in both tumor cells (Fig. 4i) and macrophages (Fig. 4j). However, comparative analysis revealed that PD-L1 expression on macrophages, unlike that on tumor cells, was consistently associated with longer PFS (Extended Data Fig. 4a,b). To address limitations due to sample size, we combined data from the Yale and UQ cohorts to enhance statistical power (Extended Data Fig. 4c,d). The pooled analysis confirmed a significant association between macrophage PD-L1 expression and PFS, while tumor cell PD-L1 remained nonsignificant. Notably, PD-L1 expression on tumor cells correlated with clinical response in only one cohort; however, this correlation was not replicated in the other, indicating inconsistency across datasets. Regression analysis further supported this trend. PD-L1 expression was strongly correlated with the mean proportions of M1 or M2 macrophages or CD14+ myeloid cells ($P < 0.001$, Spearman's rank), but not with PanCK+ tumor cells ($P = 0.17$, Spearman's rank; Extended Data Fig. 4e–h). These findings suggest that the presence of PD-L1 on macrophages could be indicative of a positive response to treatment.

## Validation of cell-type signatures using WTA deconvolution

To validate the proteomic findings, we performed cell-type deconvolution using whole transcriptomic data. We used CIBERSORTx[10] with the LM22 gene signature matrix, which profiles immune cell types enriched in the stromal region. Figure 5a shows a schematic of the deconvolution workflow applied to stromal gene expression data. Transcriptomic profiles were deconvolved into cell-type fractions, and patients were stratified by PFS at 2 years. Figure 5b demonstrates increased M2 macrophage fractions (red cells) in patients with longer PFS. To confirm this, we performed two-tailed $t$ tests comparing groups with longer and shorter PFS. Results show a substantial increase in M2 macrophages in patients with better outcomes ($P = 0.02$; Fig. 5c). These transcriptomic results are consistent with our proteomic analysis, reinforcing the role of M2 macrophages in predicting favorable responses to immunotherapy.

## Cell-to-gene signatures for resistance

We developed resistance-associated gene signatures from whole transcriptomic data in the same cohorts profiled proteomically, within the tumor compartment of advanced NSCLC. DSP-GeoMx WTA was performed across all cohorts, and one ROI was excluded from further analysis due to discrepancies across experiments (Supplementary Fig. 6). The overall workflow for gene signature development is outlined in Extended Data Fig. 5. For this purpose, we first isolated cell-type-specific genes based on CIBERSORTx and Lung Cancer Atlas (LuCA) cell-type signatures, focusing on granulocytes, vessels and proliferating tumor cells (~500 genes; Methods). Following an approach based on a previously established robust signature training framework, models were trained and tested on multiple splits of the training set to identify genes consistently associated with the highest performance, which were then included in the final resistance model. This approach selected eight genes with positive coefficients (indicative of resistance), *KRT7*, *KRT18*, *EFNA1*, *SERINC2*, *FZD6*, *CD24*, *CCND3* and *S100A9* (Fig. 6a). Higher signature scores were associated with worse outcomes across all cohorts. Using the tertile cut point established in the training cohort, high signature scores were linked to a hazard ratio (HR) of 5.3 ($P < 0.001$, two-sided log-rank test) in the Yale training cohort (Fig. 6b). Validation in the UQ and Greek cohorts confirmed these associations (HR = 2.2, $P = 0.036$; HR = 1.7, $P = 0.042$, respectively; one-sided log-rank test; Fig. 6c,d). Two-tailed log-rank tests were used in the discovery/training (Yale) cohort, whereas one-tailed log-rank tests were applied in validation cohorts, to test for effects in the same direction as observed in the discovery cohort. Multivariable analysis adjusting for clinical variables confirmed the signature's independent prognostic value (Supplementary Tables 4 and 5). However, limited heterogeneity in some covariates in the Greek cohort (which contained over 92% stage IV patients and 86% receiving first-line ICIs) and small sample size in the UQ cohort (in which some histologic subtypes were underrepresented) may limit the capacity of multivariable models to effectively

**Fig. 4 | Spatial analysis of cellular neighborhoods and immune cell interactions in the TIME. a,b,** The Voronoi diagrams for (**a**) a nonprogressed patient, illustrating the spatial distribution of cell types in the response model, and (**b**) a disease-progressed patient, illustrating the spatial distribution of cell types in the resistant model, are shown. **c,** Ten distinct cellular neighborhoods identified by the spatial interaction analysis are shown. **d,** Frequency of spatial neighborhoods within each patient sample in the Yale training cohort, stratified by progression status (nonprogressors, $n = 13$; progressors, $n = 19$). Box plots display the median (centerline) and IQR (interquartile range; 25th–75th percentile; box). Whiskers extend to the most extreme data points within 1.5× IQR. Outliers are shown as individual points. Each dot represents one patient. A total of $n = 32$ patients were analyzed using spatial proteomics, with one biological replicate (ROI) per patient. Neighborhoods were defined via spatial clustering of CODEX data. **e,** Spatial interactions between pairs of cellular constituents are shown. * indicates statistical significance at $P < 0.05$ (OLS $F$ test). **f,** Changes in the cellular constituents within neighborhoods and their association with therapy response are shown. **g,** The UMAP plot visualizes the clustering of different cell types within the tumor microenvironment, showing the distribution and grouping of cells based on their phenotypic profiles. **h,** A series of UMAP plots, each highlighting the expression levels of specific markers across the previously identified clusters is shown. Expression levels of PD-L1 on (**i**) tumor cells and (**j**) macrophages in the Yale training cohort are shown, stratified by progression status (nonprogressors, $n = 13$; progressors, $n = 19$). Violin plots show the distribution of expression values, with overlaid box plots indicating the median (centerline), IQR (25th–75th percentile; box), and whiskers extending to 1.5× IQR. Individual dots represent patients. Two-tailed $t$ tests were used to compare groups, yielding $P = 0.048$ for tumor cells and $P = 0.045$ for macrophages. A total of $n = 32$ patients were analyzed using spatial proteomics, with one biological replicate (ROI) per patient. IQR, interquartile range

control for confounding. Differential expression analysis showed upregulation of these genes in patients who progressed or died on immunotherapy (Extended Data Fig. 6a). Collectively, these genes are significantly associated with Gene Ontology (GO) biological processes related to epithelial-mesenchymal transition (EMT), regulation of cell adhesion, regulation of locomotion and regulation of epithelial cell migration ($P < 0.001$, $F$ tests; Extended Data Fig. 6b), supporting their role in tumor progression and metastasis.

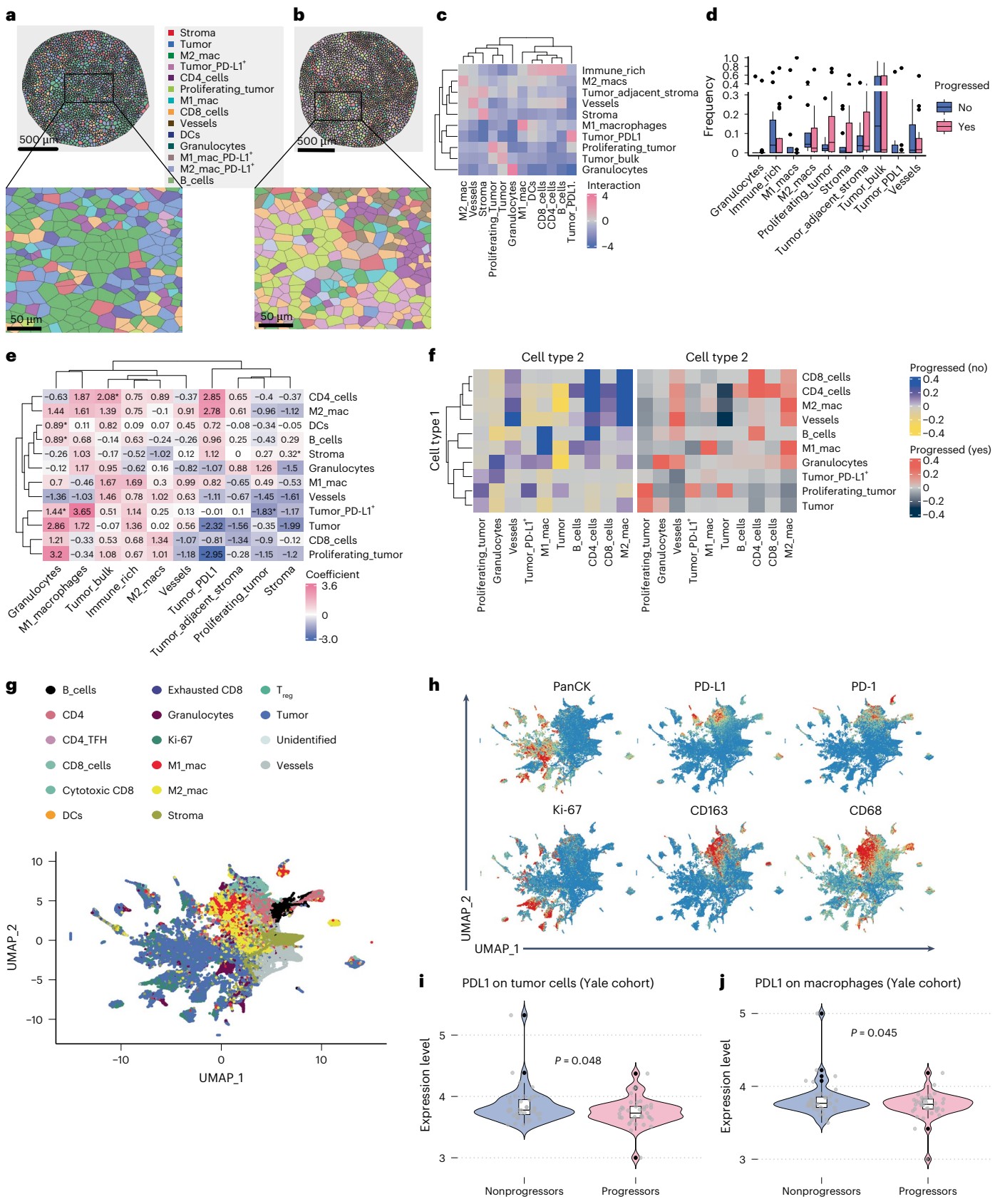

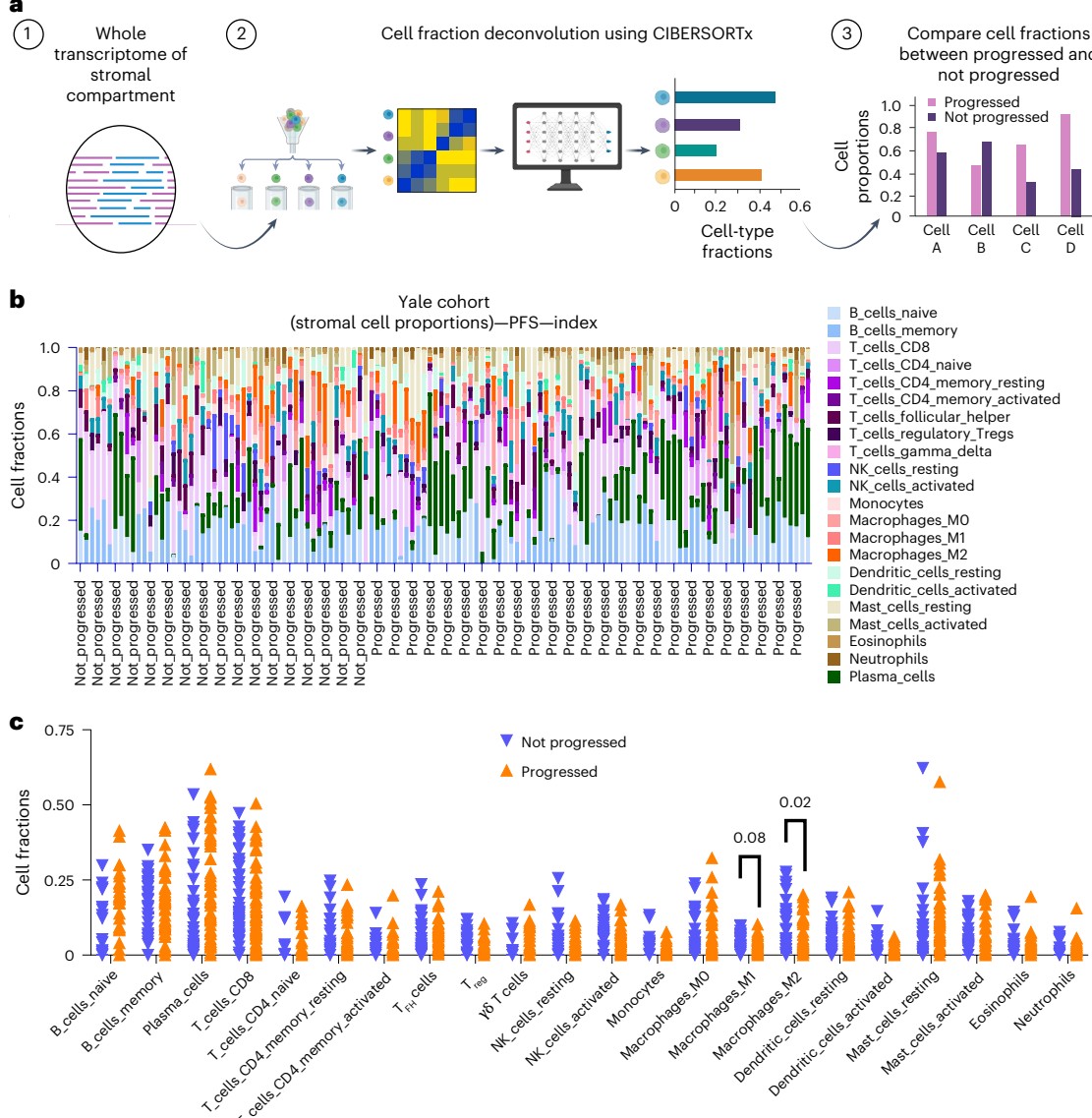

**Fig. 5 | Cell-type deconvolution of stromal compartment transcriptomic data and orthogonal validation of cell-type signatures. a**, The diagram outlines the step-by-step workflow of RNA deconvolution used to estimate cell fractions from bulk RNA data. Panel **a** was created with BioRender.com. **b**, The proportions of deconvolved stromal cells in the Yale cohort are stratified by PFS outcome, with the LM22 signature matrix from CIBERSORTx used to determine cell fractions and quantify different stromal cell types. **c**, Comparison of stromal immune cell fractions between patients who progressed (*n* = 22) and those who did not (*n* = 16)

in the Yale cohort, based on CIBERSORTx deconvolution of spatial transcriptomic data. The *x* axis represents patient groups (progressors versus nonprogressors); *y* axis represents stromal immune cell fractions. Of the 39 patients analyzed, 38 had available CD45⁺, CD68⁺ or combined stromal compartments. Statistical significance was assessed using two-tailed unpaired *t* tests without adjustment for multiple comparisons. Exact *P* values are shown above comparisons where relevant. Data points represent individual patients.

## Cell-to-gene signatures for response

We developed a response gene signature based on response cell-type models within the stromal compartment using a similar approach to the resistance gene signature training. Gene sets were derived from immune cell types using CIBERSORTx's LM22 matrix and LuCA dataset, which were used to deconvolve stroma transcriptomics and compare profiles by PFS status at 2 years. We focused on the expression of genes associated with M1 and M2 macrophages and CD4 T cells (~200 genes) to construct response models. The final model contains eight genes with negative coefficients (indicative of response), *SIGLEC1*, *TLR2*, *CXCL9*, *CD81*, *MRC1*, *CCL8*, *CCL13* and *FCGR1A* (Fig. 6e). High signature scores, defined by the upper tertile in the training cohort, were associated with improved outcomes. In the Yale training cohort, high scores were correlated with significantly better outcomes (HR = 0.22, *P* = 0.005,

two-sided log-rank test). Validation in the UQ cohort confirmed the association (HR = 0.38, *P* = 0.034, one-sided log-rank test; Fig. 6f,g) and the Greek cohort showed a consistent trend (HR = 0.56, *P* = 0.041; Fig. 6h). Multivariable analysis, adjusting for clinical covariates, confirmed the signature's independent prognostic value in both validation cohorts (Supplementary Tables 6 and 7). Differential expression analysis showed these genes were significantly upregulated in nonprogressors (Extended Data Fig. 7a) and were enriched in immune-related processes, including leukocyte and lymphocyte activation (Extended Data Fig. 7b), supporting their role in mediating antitumor immune responses.

## Cell-to-gene signature association with 5-year PFS and OS

We further assessed the association of the gene models with additional clinical endpoints, including 5-year PFS, 5-year OS and 2-year OS. For

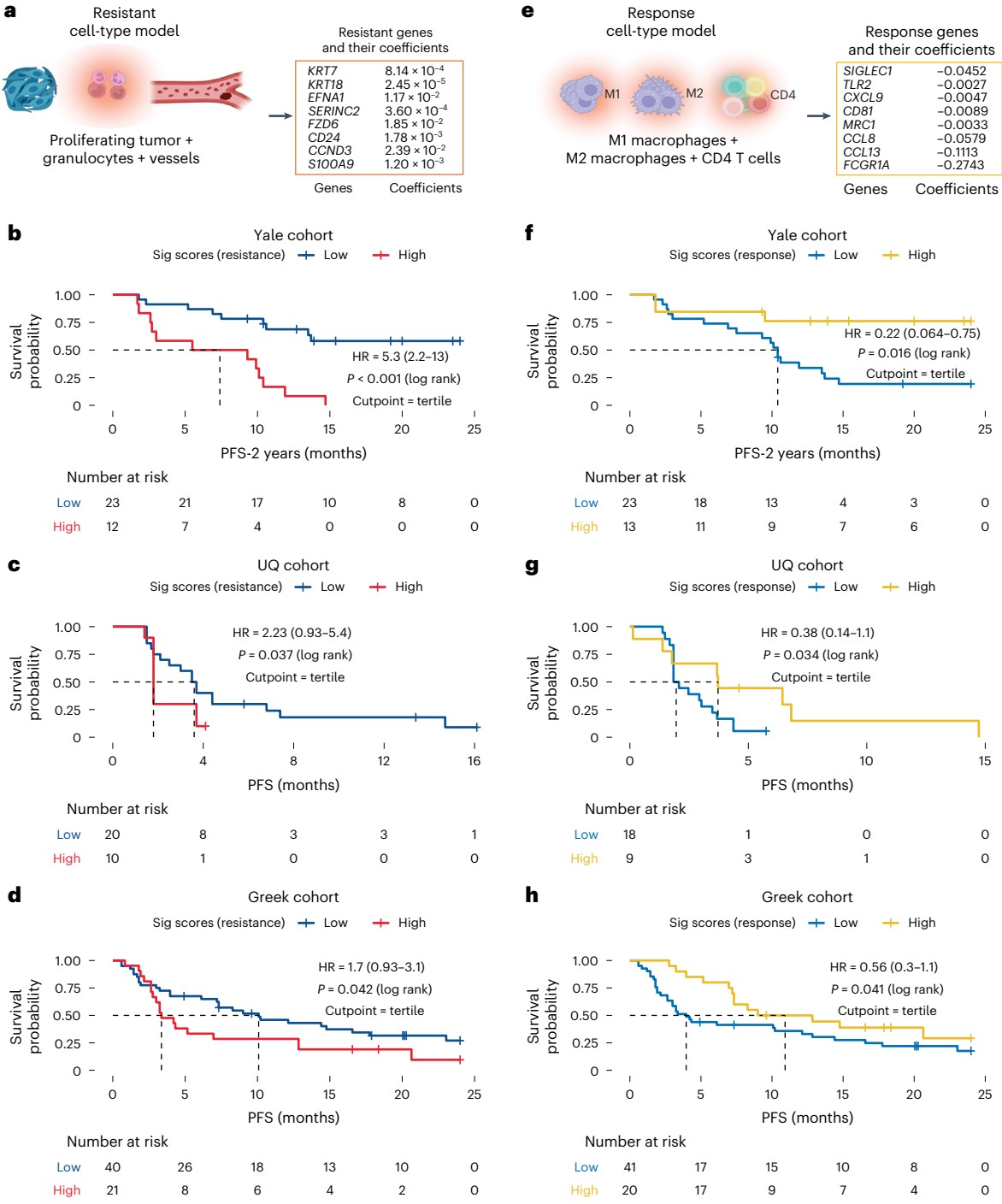

**Fig. 6 | Cell-to-gene signatures using DSP-GeoMx WTA data for resistance and response. a**, Genes included in the resistance signature model, derived from their enriched cell types. The listed genes, along with their signature coefficients, indicate the magnitude and direction of each gene's contribution to the prediction of PFS. **b–d**, KM plots showing PFS stratified by resistance gene signature scores in the tumor compartment of the Yale (**b**), UQ (**c**) and Greek (**d**) cohorts. Patients were divided into high and low groups using the upper tertile as the cut point. Statistical significance was assessed using two-tailed log-rank tests. No adjustments were made for multiple comparisons. The number at risk is shown below each curve. **e**, Genes included in the response signature model, derived from their enriched cell types. The listed genes, along

with their signature coefficient, indicate the magnitude and direction of each gene's contribution to the prediction of PFS. Panels **a** and **e** were created with BioRender.com. **f–h**, KM plots validating the response gene signature model in the stromal compartment of the Yale (**f**), UQ (**g**) and Greek (**h**) cohorts. Patients were dichotomized into high and low groups using the upper tertile of the response signature score as the cut point. Two-tailed log-rank tests were used in the discovery cohort (Yale), while one-tailed log-rank tests were used in the validation cohorts (Greek and UQ), with the direction of effect prespecified based on the discovery data. No adjustments were made for multiple comparisons. The number at risk is indicated below each curve.

the resistant model, we applied a tertile cut point for 5-year PFS and median cutpoints for OS endpoints, all established in the Yale training cohort (Extended Data Fig. 8a–f) and used consistently across

UQ (Extended Data Fig. 9a,b) and Greek (Extended Data Fig. 10a–f) validation cohorts. In the Yale cohort, both resistance and response gene models were significantly associated with longer 5-year PFS. The

resistance model was also significantly associated with both 5-year and 2-year OS, while the response model showed only a trend with *P* values of 0.06 (HR = 0.34) for OS at 5 years and 0.09 (HR = 0.37) for OS at 2 years. These findings suggest that the resistance gene model more robustly predicts both PFS and OS, whereas the response gene model appears specific to PFS. This underscores the importance of selecting appropriate clinical endpoints when evaluating the efficacy of these models. The stronger association of the resistance model with OS highlights its potential utility in identifying higher-risk patients and guiding long-term treatment strategies. In the UQ cohort, limited follow-up time restricted the OS analysis to months. Although the resistance model showed a trend toward worse PFS (HR = 2.3, *P* = 0.06), significance was not reached. The response model showed no validation, likely due to insufficient follow-up time. In the Greek cohort, high resistance scores were significantly associated with worse 5-year PFS (HR = 1.8, *P* = 0.029). OS showed a similar trend, with HR = 1.6 (*P* = 0.06) at 5 years and HR = 1.5 (*P* = 0.14) at 2 years. For the response model, higher scores trended with improved outcomes for PFS at 5 years (HR = 0.61, *P* = 0.06), OS at 5 years (HR = 0.67, *P* = 0.11) and OS at 2 years (HR = 0.62, *P* = 0.12). The consistent trends across different endpoints support the prognostic potential of both models.

## Discussion

Our study presents a spatial multi-omic framework for developing biomarkers associated with PD-1-based immunotherapies in solid tumors. In this study, rather than directly generating protein-based signatures, we transformed protein expression into phenotypic cell states and then derived cell-type signatures linked to immunotherapy outcomes. Genes were subsequently extracted from these cell types to construct transcriptomic signatures associated with immunotherapy outcomes. The rationale behind this approach lies in the difficulty of directly combining proteomic and transcriptomic data, which is challenging due to the nonlinear dependency between protein and RNA expression. Incorporating spatial context further strengthens this approach by enabling the identification of not only relevant cell types and genes but also their localization within the TIME. Spatial coordinates embedded in outcome models provide a layer of evidence, capturing interactions shaped by tissue architecture.

We first developed resistance and response cell-type models from spatial proteomic data. These models identified distinct cell-type distributions associated with PFS and enabled the creation of a hierarchical map of gene signatures within the same spatial locations. The resistance signature (comprising proliferating tumor cells, granulocytes and vessels) was enriched in the tumor compartment, forming a resistant niche that promotes cancer progression. In contrast, the response signature, enriched in M1 and M2 macrophages and CD4 T cells, was prominent in the stroma, highlighting the importance of the stromal immune environment in mediating treatment responses. To validate these findings, we conducted orthogonal validation using transcriptomic data and the CIBERSORTx deconvolution, confirming a higher abundance of M2 macrophages in patients with longer PFS, consistent with the response stromal signature. The spatial analysis proved pivotal in clarifying the role of PD-L1 expression on macrophages. Spatial mapping of PD-L1 expression further clarified its role, showing PD-L1-positive macrophages contribute to a favorable TIME and enhance PD-1-based immunotherapy efficacy. Current companion diagnostics in NSCLC, such as pembrolizumab eligibility, rely on the tumor proportion score (TPS), which measures PD-L1 expression on tumor cells alone[11–13]. In contrast, the combined positive score (CPS), which includes immune cells, has shown better correlation with treatment response, aligning with our findings[14,15]. Furthermore, research in refs. [16,17] showed that PD-L1 expression on macrophages enhances CD8[+] T cell proliferation and cytotoxicity, supporting the use of CPS. Neighborhood and cell interaction analysis revealed key spatial dynamics underlying therapeutic outcomes. Patients with disease progression had elevated communities of proliferating tumor cells, granulocytes and vessels, indicating their collaborative roles in tumor proliferation and angiogenesis. In contrast, interactions among M1 macrophages, B cells and CD4 T cells in responders point to effective immune regulation. Increased interactions between M2 macrophages and both CD4 and CD8 T cells in nonprogressors suggest a role in favorable outcomes. Macrophage plasticity, specifically repolarization from M2 to M1 phenotype, can enhance antitumor immunity, making macrophages attractive therapeutic targets[18–20]. A recent study shows that targeting MS4A4A on M2 macrophages can restore CD8[+] T-cell-mediated antitumor immunity and improve anti-PD-1 therapy efficacy[21]. Our findings support the critical role of M2 macrophages in modulating the TIME and their potential to enhance immunotherapy responses, supporting the value of comprehensive markers such as CPS.

Building upon the cell-type signatures, we developed gene-based signatures using a cell-to-gene modeling approach. For the resistance signature, genes such as *KRT7*, *KRT18* and *FZD6* are linked to EMT, a process enabling tumor invasion and metastasis, and contribute to chemoresistance and reduced immunotherapy response[22]. *EFNA1* and *FZD6* also promote angiogenesis, supporting tumor growth and immune evasion by creating hypoxic environments[23]. *S100A9* and *CD24* are associated with immune suppression. *S100A9* (and its partner, *S100A8*) activates endothelial cells via *TLR4* and *RAGE*, producing proinflammatory cytokines and promoting a protumor microenvironment[24], while *CD24* mediates immune evasion[25] via *SIGLEC-10* (ref. [26]). *CCND3* and *SERINC2* promote cell cycle progression and proliferation, with *CCND3* regulating the G1/S transition[27], and *SERINC2* contributing to lipid biosynthesis and cancer cell survival[28].

The response signature includes genes such as *SIGLEC1* and *CXCL9*, which mediate immune activation. *SIGLEC1* is crucial for macrophage differentiation and immune responses[29], while *CXCL9* attracts immune cells and promotes T cell infiltration[30]. *TLR2*, highly expressed on M1 macrophages, supports innate immune activation[31]. Elevated *CD81* on CD4 T cells enhances migration and adhesion, aiding antitumor responses[32]. *MRC1* (*CD206*), a marker of M2 macrophages[33], typically associated with immune suppression, may adopt a more proinflammatory role when co-expressing high PD-L1, potentially enhancing the immunotherapy response. *CCL8* and *CCL13* facilitate immune cell recruitment to inflammation sites[34], potentially improving immunotherapy outcomes. *FCGR1A*, expressed on macrophages, enables antibody-dependent phagocytosis[35], supporting tumor clearance in the presence of opsonizing antibodies. These resistance and response gene signatures collectively highlight distinct spatial patterns. It is noteworthy that the resistance signature primarily originates from the tumor compartment, while the response signature is specific to the stromal compartment. The tumor compartment fosters resistance through EMT, cell migration and angiogenesis, while the stromal compartment contributes to therapeutic response via immune cell activation and differentiation. These spatial gene signatures offer valuable insights into mechanisms underlying immunotherapy outcomes in advanced NSCLC.

While our findings provide strong support for the utility of multiomics profiling in predicting outcomes, we acknowledge several limitations. Notably, differences in the proportion of disease stages, treatment regimens and line of immunotherapy in the validation cohorts, constrain the interpretability of multivariable results. The UQ cohort had limited representation of certain subtypes. The Greek cohort predominantly included stage IV patients treated with first-line immunotherapy, intentionally selected to test our signatures in this setting. We acknowledge that the broad generalizability of our signatures in highly heterogeneous cohorts requires further testing. Furthermore, we acknowledge that, due to the reduced number of patients in certain covariate subgroups in the validation cohorts, the multivariable analysis may not be able to quantify the effects of all covariates with high accuracy. Therefore, an accurate determination of the effect sizes

would benefit from testing on larger cohorts. Finally, PD-L1 expression, tumor mutation burden and performance status were not consistently available and thus excluded from our models.

Due to the small number of patients who received only first-line immunotherapy in the UQ validation cohort, we included patients treated with first, second and third line or beyond immunotherapy, introducing some heterogeneity. The ability of our models to achieve statistical significance in this setting adds credibility to their broad potential clinical utility, although, as noted above, the generalizability of the signatures as prognostic models requires further validation in more heterogeneous datasets. Future retrospective studies with extended follow-up and prospective validation will be essential to strengthen clinical relevance. Ultimately, the development of cell-to-gene signatures, augmented by spatial information, represents a substantial step forward in personalized medicine for NSCLC.

## Online content

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

[1]Department of Pathology, Yale University School of Medicine, New Haven, CT, USA. [2]Frazer Institute, Faculty of Medicine, The University of Queensland, Brisbane, Queensland, Australia. [3]NEC Laboratories America, Princeton Office, Princeton, NJ, USA. [4]Department of Molecular Biophysics and Biochemistry, Program in Computational Biology and Bioinformatics, Yale University, New Haven, CT, USA. [5]Bioinformatics Division, The Walter and Eliza Hall Institute of Medical Research, Parkville, Victoria, Australia. [6]Department of Medical Biology, Faculty of Medicine, Dentistry and Health Sciences, University of Melbourne, Parkville, Victoria, Australia. [7]Queensland University of Technology, Centre for Genomics and Personalized Health, School of Biomedical Sciences, Brisbane, Queensland, Australia. [8]Oncology Unit, Department of Medicine, University of Athens, Athens, Greece. [9]Yale Cancer Center, New Haven, CT, USA. [10]These authors contributed equally: Thazin N. Aung, James Monkman, Jonathan Warrell. ✉e-mail: arutha.kulasinghe@uq.edu.au; david.rimm@yale.edu

## Methods

### Study population

We studied three independent cohorts of patients with NSCLC who were treated with PD-1-based immunotherapies in the advanced or metastatic setting. The clinical characteristics of the cohorts are presented in Supplementary Tables 1 and 2. The first cohort, which served as the training cohort, was obtained from the Yale Cancer Center (New Haven). This cohort comprises 113 tissue samples collected between 2012 and 2019 (Fig. 1a). Tissue microarray (TMA) master blocks were created using nonadjacent tumor cores, each with a diameter of 0.6 mm, from each patient's biopsy. To represent tumor heterogeneity, 5 µm-thick cuts from four independent blocks per TMA, each containing a different core (ROI) from the same patient tumor, were used for transcriptomic profiling using the DSP-GeoMx WTA or CTA (cancer transcriptome atlas, Bruker Spatial Biology, US) platforms and single cores were used for proteomic profiling using the PhenoCycler fusion (PCF) platform (Akoya Biosciences, US). The GeoMx WTA platform and two ROIs per TMA from the same patient tumor were used for proteomic profiling using the PCF platform. These samples were collected and used under the authorization of the Yale Human Investigation Committee (protocol 9505008219), with assurances filed with and approved by the U.S. Department of Health and Human Services. The Yale cohort TMA is designated as YTMA-471. The second cohort, used as a validation cohort, was obtained from the UQ. TMAs were constructed from resected NSCLC tissue specimens from patients who subsequently recurred and were treated with PD-1-based immunotherapies in the advanced or metastatic setting. This cohort contained 42 tissue samples collected between 2009 and 2018 (Fig. 1a) and is designated as UQTMA-4301. The study received ethical approval from the Queensland University of Technology Human Research Ethics Committee (2000000494) and was ratified by the UQ. The third cohort, serving as a second validation dataset, was obtained from Sotiria General Hospital, Medical School, National and Kapodistrian University of Athens, Greece. It comprised 79 patients who received PD-1-based therapy between 2019 and 2023 and was used to validate the generalizability of the findings from the Yale training cohort. The study was conducted under HIC protocol 16760/23-06-2023. The Greek cohort TMAs are YTMA-642 + YTMA-709. Written informed consent or waiver of consent was obtained from all participants before enrollment. A schematic of the workflow is shown in Fig. 1b. Clinical data are provided in Supplementary Data 1 and 2.

### Study design

As stated above, the Yale cohort served as a training set, while the UQ cohort and the University of Athens (Greek cohort) served as validation sets. The training cohort facilitated the development of biomarkers associated with immunotherapy outcomes in machine learning models, while the validation cohorts provided independent data to assess the generalizability of these models. Treatment response was classified according to Response Evaluation Criteria In Solid Tumors version 1.1 (Complete Response, Partial Response, Stable Disease and Progressive Disease), and PFS was recorded for all cohorts. To minimize biases from prior therapies, the study focused solely on patients receiving first-line immunotherapy. As a result, the Yale cohort was narrowed down to 41 patients. The clinical endpoint was set to 2-year PFS, consistent with prior real-world and clinical trial data supporting this timeframe as a clinically meaningful surrogate for durable benefit in advanced/metastatic NSCLC. Major immunotherapy trials that led to regulatory approval, such as KEYNOTE-189 and KEYNOTE-407, implemented a 2-year cap on treatment duration[2,11]. In real-world clinical practice, treatment duration beyond two years was common due to the absence of data defining optimal therapy length. However, recent retrospective evidence from over 1,000 patients showed no significant OS benefit beyond two years[36]. Furthermore, approximately 20% of patients in that study discontinued treatment at two years despite the

absence of progression. Taken together with the toxicity and financial burdens of prolonged therapy, these findings support a shift toward 2-year treatment as a clinical standard. Therefore, adopting 2-year PFS as an endpoint enhances the real-world relevance and interpretability of our findings[37]. Exclusion criteria for each cohort are depicted in the CONSORT diagrams (Fig. 1a,b). A schematic of data acquisition and analysis workflow is shown in Fig. 1b. We began with data acquisition using the PCF platform and validated findings using the DSP-GeoMx platform. The final analysis focused on findings based on cell types identified through the PCF-derived proteomic data.

### Spatial proteomic profiling for Yale and UQ NSCLC cohorts

The CODEX staining method was used for Yale, UQ and Greek cohorts, using the Phenocycler Fusion platform. Staining and imaging were performed by the Akoya Biosciences Spatial Tissue Exploration Program (STEP) following the manufacturer's instructions[38]. Formalin-fixed, paraffin-embedded TMA sections were mounted on SuperFrost charged slides, rehydrated and subjected to heat-induced epitope retrieval. A 26-antibody cocktail (190 µl per section) was incubated for 3 h at room temperature in a humidity chamber. The list of antibodies is provided in Supplementary Table 3. This was followed by several cycles of washing and fixation. Details on antibody barcoding, dilution and imaging cycles are provided in Supplementary Data 3. Imaging was performed using the PCF platform and registered qptiff files were exported for image analysis.

### Image analysis

Images were imported into QuPath (version 0.4.2)[39] and segmented using the Cellpose plugin (version 2.0)[40] on the DAPI2 channel with the nuclear pretrained model ('.cellExpansion = 4 µm' and '.cellConstrainScale = 1.5'). Segmentation quality was visually inspected, and TMA cores with poor quality, such as those that were fragmented, folded, necrotic, or exhibiting high nonspecific fluorescence, were excluded. An artificial neural network pixel classifier was trained on PanCK (pan-cytokeratin) signals to define a tumor/stroma mask, capturing PanCK+ pixels as tumor. Only tumor nests larger than 100 µm² were annotated as 'tumor'. Cell metrics, including universally unique identifier (UUID) codes, spatial coordinates, nuclear size and median cell expression per channel, were exported for analysis in Python. Cell classifications from subsequent unsupervised clustering were re-imported into QuPath and matched by their UUIDs for visual inspection and quality control. A detailed image analysis workflow is illustrated in Extended Data Fig. 1a. Raw fluorescence intensity data (16-bit) underwent arcsinh transformation to normalize and reduce skewness, followed by marker scaling across cells. CD45 was used to illustrate preprocessing effectiveness (Extended Data Fig. 1b).

### Data integration, clustering and phenotyping workflow

Expression matrices and cell metadata were imported into Anndata for quality control, preprocessing, clustering and phenotyping[41–43] in Python 3.10. Data were first subset into canonical lineage markers (CD45, CD3e, CD4, CD8, CD20, CD14, CD68, CD11b, CD11c, CD31, CD34, SMA, Vimentin, E-cadherin, PanCK, nuclear size). Each slide was preprocessed individually by arcsinh transform (cofactor 150), marker-wise scaling, and cell-wise normalization, following recommended CODEX workflows[44]. Preprocessed data were concatenated and integrated using Scanpy's Harmonypy (version 0.0.5)[45] by slide, and adjusted principal components were used for clustering via Phenograph (version 1.5.7)[46]. Multiple Leiden resolutions (0.1–0.5) were tested across $k = 15–30$ and $r = 0.2$, with $k = 15$ empirically chosen to best resolve canonical cell types. Eighteen clusters were manually annotated to identify CD4, CD8, B cells, granulocytes, macrophages, dendritic cells (DCs), tumor cells, stromal cells, vessels and unidentified cells. Further classification in QuPath defined functional subsets, including CD4 $T_{FH}$ (PD-1+, ICOS+), $T_{reg}$ (FOXP3+), exhausted CD8 (PD-1+),

cytotoxic CD8 (granzyme B[+]), M2 macrophage (CD163[+]), proliferating tumor (PanCK[+]Ki67[+]) and PD-L1 tumor (PD-L1[+]; Fig. 2a), using an object classifier trained on representative images. Labeled cells were then merged into 14 final cell types (Fig. 2b). For the analysis of PD-L1 expression in progressors and nonprogressors in tumor cells and macrophages, PanCK[+] cells above the median were used to compute mean tumor PD-L1 per patient. Similarly, cells expressing CD68 and CD163 above the threshold were used to quantify macrophage PD-L1.

## Spatial analysis

Cell proportions were calculated by normalizing the number of each cell type within a TMA core or tumor/stromal region to the total number of cells within that region, yielding cell percentages. Spatial interactions were analyzed using the 'spatial_interaction' function in Scimap (https://github.com/labsyspharm/scimap) with the radius method (radius = 30 μm, $k$-nearest neighbor = 10) for all pairwise combinations. Group means were compared between PFS groups. Cellular neighborhoods were defined using the Neighborhood Identification Pipeline (https://github.com/nolanlab/NeighborhoodCoordination)[9]. For each cell, the types of its ten nearest neighbors were recorded as features, which were then clustered into ten groups using a $k$-nearest neighbor unsupervised algorithm. The selection of ten neighbors and ten clusters was guided by prior literature[9,47]. Each cell was assigned to a 'neighborhood', based on the most frequent nearby cell types, and these clusters were manually annotated by their dominant phenotype. Differential enrichment of cell types across neighborhoods was assessed using an ordinary least squares linear model from the same pipeline[9], measuring fold change significance based on binary PFS outcome.

## DSP-GeoMx WTA for the Yale and Greek NSCLC cohorts

For the Yale cohort, TMA slides were processed using the DSP-GeoMx manual slide preparation protocol (MAN-10150-01) as described in our prior study[5]. Briefly, four different ROI containing distinct tumor areas of the same tumors were used to enhance reproducibility. Antigen retrieval of the formalin-fixed, paraffin-embedded tissue was conducted for 20 min, followed by deparaffinization and rehydration. The slides were then exposed to proteinase K for 20 min, after which RNA probes were applied to the tissues for in situ hybridization. The following day, stringent washes were performed to remove off-target probes. To delineate areas within each ROI, morphology markers were used for each AOI−CD68 for macrophage, CD45 for leukocyte, PanCK for tumor compartments and SYTO 13 for nuclear staining. The slides were incubated, washed and loaded onto the DSP-GeoMx instrument. Scanning, AOI selection and probe collection were performed as per the user manual (MAN-10088-03). Each AOI representing a compartment, such as tumor region, macrophage region or leukocyte region from a patient tumor core was collected in a 96-well plate. Next, a GeoMx-next-generation sequencing (NGS) readout library was prepared according to the user manual (MAN-10117-01). Sequencing was performed by Yale Center for Genome Analysis, and the sequencing data were processed with the GeoMx-NGS Pipeline (MAN-10153-01), converting FASTQ files to digital count conversion (.dcc) files for read counts. Quality control was conducted on each ROI to ensure accurate analysis of the approximately 18,000 genes targeted using the WTA panel.

## DSP-GeoMx CTA processing for the UQ NSCLC cohort

For the UQ cohort, pretreatment TMA slides were processed using the DSP-GeoMx CTA manual slide preparation protocol. This methodology was adapted from ref. 5, which profiles approximately 1,800 genes (Cancer Transcriptome Atlas Panel). After antigen retrieval and deparaffinization, proteinase K treatment was performed, followed by in situ hybridization using CTA RNA probes. The stringent washes were then performed to remove off-target probes. CD3, CD68 and PanCK morphology markers were used to identify tumor AOIs (PanCK[+]) and stromal (PanCK[−]) regions. The slides were loaded onto the DSP-GeoMx

instrument, and scanning, AOI selection, and probe collection were conducted per the DSP-GeoMx user manual. Each AOI representing a compartment, tumor or stroma, was collected into a 96-well plate. A GeoMx-NGS readout library was prepared following the DSP-GeoMx user manual (MAN-10117-01). PCR reactions were pooled into two mixtures for the tumor and stromal compartments. Sequencing was performed by the Australian Genome Research Facility, and data were processed using the GeoMx-NGS Pipeline (MAN-10153-01) to convert FASTQ files to digital counts conversion (.dcc) files for read counts. Quality control was performed according to preset thresholds to ensure accurate and high-quality profiling.

## Univariable analysis of cell-type fractions associated with treatment outcome

We conducted univariable analysis to assess the association between cell-type fractions and treatment outcomes for 14 cell types in both tumor and stromal compartments. Cell fractions were derived using the PCF platform. HRs and two-sided log-rank test $P$ values were calculated using Cox proportional hazards models to evaluate the impact of each cell type on 2-year PFS. BH correction was applied for multiple hypothesis testing.

## Computational pipeline for the development of cell-type signatures

We developed cell-type signatures for both tumor and stromal compartments using a robust signature training framework based on a prior study[5]. The input consists of matrices $M_{ij}^{tumor}$ and $M_{ij}^{stroma}$ containing cell fractions for each patient $i$ and cell-type $j$ in tumor and stroma compartments, respectively, where there are 14 distinct cell types across tumor and stromal compartments. For each patient, we have PFS data (time in months and an event indicator: 1 = progression/death and 0 = alive/progression free). For each compartment, we generated $n$ splits of the training data into tenfolds, generating $M$ LASSO-penalized Cox models. For each split, tenfold cross-validation was used to select penalty parameter $\lambda$, and a LASSO model was trained using the optimal penalty for that split by pooling all tenfolds. We set lower.limits = 0 in R glmnet (version 4.1-9) for resistance models (positive nonzero coefficients) and upper.limits = 0 for response models (negative nonzero coefficients). This yielded $M$ coefficient sets $\beta_{m,j}$, where $\beta_{m,j}$ is the coefficient of cell type $j$ in model $m$. Cell types with a nonzero coefficient in at least $t$ of the models were retained. A final unpenalized Cox model was fit using these selected cell types to obtain coefficients $\beta_j^{final}$. The final

signature score for patient $i$ was calculated as $S_i^{final} = \Sigma_j \beta_j^{final} M_{ij}$. Hence,

the final signature is a Cox regression model trained on LASSO-selected cell types. This pipeline was applied to both tumor and stromal compartments to derive resistance and response signatures, where we set $M = 50$ for stroma and $M = 100$ for tumor, since we observed that these values were sufficient to identify a stable set of nonzero predictors for each compartment, and the threshold $t$ was set to the maximum value such that each signature contained at least three cell types.

## Independent validation of compartment-specific cell-type signatures

We computed the final signature scores for resistance and response models in both tumor and stromal compartments. Each patient received a final score, indicating their predicted risk of progression based on the specific compartment and cell-type signature. We assessed out-of-sample accuracy by evaluating HR and survival probabilities in the validation cohorts. In resistance models, positive coefficients predicted increased risk HR > 1.0; in response models, negative coefficients predicted reduced risk HR < 1.0. Statistical significance of HRs was assessed using Cox proportional hazards models. $P$ values were calculated using a two-tailed log-rank test by binarizing the scores at the median and refitting Cox models.

## Cell-type deconvolution using DSP-GeoMx- data for validation of spatial proteomic data

To validate the proteomically identified cell types, we analyzed gene expression data generated from the stromal compartment using the DSP software (version 0.4). The quantile-normalized matrix was deconvolved with CIBERSORTx, using the leukocyte gene signature matrix (LM22), which defines reference gene expression profiles for 22 immune cell types. This enabled estimation of stromal cell fractions for each patient. The deconvolution model solved for the cell fractions $F_{ji}$ using: $B \times F_i \approx M_i^{\text{DSPGeoMx}}$, where $B$ is the LM22 signature matrix, $F_{ji}$ is the fraction of cells belonging to cell type $j$ in patient $i$, and $M_i^{\text{DSPGeoMx}}$ is the stromal compartment matrix for patient $i$. We compared cell fraction distributions between patients with different PFS outcomes (alive versus dead) using two-sided $t$ tests. This analysis identified stromal cell types enriched in either group. By cross-validating our compartment-specific signatures against DSP-derived CIBERSORTx results, we confirmed the consistency and predictive relevance of the spatial proteomic cell-type features.

## Computational pipeline for the development of gene signatures from cell-type signatures (cell-to-gene signatures)

We developed compartment-specific gene signatures based on the proteomics-derived cell-type signatures using a robust signature training framework based on the approach developed in a prior study[5]. Gene expression reference profiles were obtained from CIBERSORTx, and a public lung cancer scRNA-seq dataset (LuCA) focusing on CD4 T cells, macrophages, granulocytes and malignant cells. Reference gene lists for both models are provided in Supplementary Data 4. Using the mastR (version 1.8.0) R package[48], 892,296 cells from LuCA were aggregated into 2,215 pseudobulk samples (>50 cells each), spanning 14 cell types. Marker genes were identified using differential expression and a ranked product permutation test with default parameters. We constructed matrices $M_{ij}^{\text{tumor}}$ and $M_{ij}^{\text{stroma}}$ containing gene expression counts for each patient $i$ and gene $j$ in tumor and stroma compartments, respectively, where genes were restricted to those associated with cell types occurring in the associated proteomics cell-type signatures. Each patient had PFS data (time in months and event indicator−1 = progression/ death, 0 = progression free). For each compartment (tumor and stroma) and each cell-type gene set, we generated 50 data splits, each split 80/20 for training/testing. In each training split, differentially expressed genes between progression-free and nonprogression-free patients ($P < 0.05$, negative binomial exact test) were identified. For each split, 100 LASSO Cox regression models were optimized with different random seeds, by first using tenfold cross-validation on the current training split to select the penalty $\lambda$, and then fitting a final LASSO Cox regression model with the best value of $\lambda$ using the current training set. Any gene appearing in at least one of these models was retained; a signature for the current data split was then trained by fitting an unpenalized Cox regression model on the current training split restricted to the selected genes, and the HR of the signature was estimated on the current test set. To constrain coefficient directions, we set lower.limits = 0 for resistance models (positive nonzero coefficients) and upper.limits = 0 for response models (negative nonzero coefficients). Each data split $m$ thus produced a coefficient set $\beta_{m,j}$ and an associated HR estimate. Genes with nonzero coefficients in at least 0.05 of the models having HR > 1.5 (resistance) or HR < 0.7 (response) were selected for inclusion in the final signatures. These thresholds were set so that the number of genes included was at most ten. Final Cox regression models were then refit without the penalty using the entire discovery cohort (not split into train/test), and only the selected genes, to derive the final signature coefficients, $\beta_j^{\text{final}}$. The final signature score for patient $i$ was calculated as $S_i^{\text{final}} = \Sigma_j \beta_j^{\text{final}} M_{ij}$. For training the resistance gene model, we used genes from cell types in the resistance cell-type signature models (for example, proliferating tumor,

granulocytes and vessels), and for training the response gene model, we used genes from the cell types in the response cell-type signature models (for example, M1 macrophages, M2 macrophages and CD4 T cells). This approach was applied uniformly across tumor and stromal compartments to define gene-level resistance and response signatures derived from cell-type-level proteomic signatures.

## Independent validation of cell-to-gene signatures

To validate the gene signatures, we calculated gene expression matrices for tumor and stromal compartments in both the UQ cohort and the Greek cohort. Sequencing and data processing for the UQ cohort were performed by the UQ research team. For the Greek cohort, sequencing and data preprocessing were conducted by the Yale team. Each matrix contained expression counts per patient and gene in the respective compartments. We computed final resistance and response signature scores for both tumor and stromal compartments, with each score indicating the predicted risk of progression. Out-of-sample performance was evaluated using HRs of the signature scores evaluated in the validation cohorts. Genes with positive coefficients in the resistance signature predicted increased risk HR > 1.0, while genes with negative coefficients in the response signature predicted reduced risk HR < 1.0. Statistical significance was assessed using Cox proportional hazards models, with two-sided log-rank test $P$ values derived from each model to test the out-of-sample accuracy of our signatures. We also conducted a multivariable analysis adjusting for clinical factors when available (age, sex, disease stage, prior treatment, type of immunotherapy, line of immunotherapy, histology and smoking status). Cox multivariable regression and Chi-square tests (Supplementary Tables 4–7) were used to evaluate associations between gene signatures and outcomes. This validation step is designed to test the generalizability of the spatially defined compartment-specific gene signatures.

## Statistics and reproducibility

No statistical method was used to predetermine sample size. One ROI was excluded from transcriptomic analysis due to pre-identified operator inconsistencies, as documented in Supplementary Fig. 6. The remaining data were included in all analyses. Samples were not prospectively randomized. The investigators were not blinded to allocation during experiments and outcome assessment.

## Reporting summary

Further information on research design is available in the Nature Portfolio Reporting Summary linked to this article.

## Data availability

Raw and processed DSP-GeoMx WTA RNA sequencing data from the Yale discovery cohort and the Greek validation cohorts are available under accession number GSE271689. DSP-CTA raw RNA sequencing data of the UQ validation cohort can be assessed via GSE221733. Single-cell Lung Cancer Atlas (LuCA) data is available via https://luca.icbi.at/ on Zenodo (https://doi.org/10.5281/zenodo.7227571)[49]. CIBERSORTx (http://cibersort.stanford.edu/) leukocyte gene signature matrix (LM22) is available in ref. 10. Source data are provided with this paper.

## Code availability

Custom analysis scripts used in this study are available at https://github.com/tznaung/NSCLC_SpatialOmics and https://doi.org/10.6084/m9.figshare.29944619.v1 (ref. 50).

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

## Acknowledgements

This work was supported by the Yale Cancer Center CCSG (grant P30CA016359) and the Yale SPORE in Lung Cancer (P50CA196530) (to D.L.R.). We thank the Yale Center for Genome Analysis for the use of its high-performance computing resources, funded by the NIH (grant 1S10OD030363-01A1). T.N.A. was supported by the Robert E. Leet and Clara Guthrie Patterson Trust Mentored Research Award, the Tower Cancer Research Foundation, and the Lion Heart Research Foundation, all through the Yale School of Medicine. A.K. is supported by the MRFF METASPATIAL Study (2031100), Cure Cancer and the Princess Alexandra Research Foundation (PARF). The funders had no role in study design, data collection and analysis, decision to publish or preparation of the manuscript.

## Author contributions

T.N.A., J.M., J.W., D.L.R. and A.K. designed the study. T.N.A., J.M. and J.W. performed statistical analyses. T.N.A., J.M. and M.M. performed the DSP-GeoMx WTA and CTA experiments. T.N.A., J.M., J.W., I.V., K.M.B., N.G., I.P.T., C.W.T., A.I.F., K.O.B., K.A.S., K.S., R.S.H., A.K. and D.L.R. provided administrative, technical and material support. T.N.A. and J.W. drafted the manuscript. T.N.A., J.M., J.W., K.O.B., K.A.S., R.S.H., A.K., D.L.R., K.M.B., N.G., I.P.T., C.W.T., I.V., A.I.F. and M.M. contributed to writing the original manuscript. D.L.R., A.K. and J.W. supervised the study. T.N.A., J.M., J.W., A.K. and D.L.R. had access to all of the data in the study and took responsibility for the integrity of the data and accuracy of the analyses.

## Competing interests

D.L.R. reports research support from AbbVie, Cepheid, Navigate BioPharma, NextCure, Konica/Minota, Regeneron and Leica/Danaher; instrument support from Akoya; major honoraria from consultative activities with Cell Signaling Technology, Cepheid, Nucleai and Danaher; and minor honoraria from consultative activities with AstraZeneca, Immunogen, PAIGE.AI, Regeneron, Roche and Sanofi. A.K. is supported by Cure Cancer and the Passe and Williams Foundation and is an advisor for the European Spatial Biology Company, Omapix Solutions, Predxbio, Molecular Instruments and Visiopharm. R.S.H. reports research support from AstraZeneca, Eli Lilly and Company, Genentech/Roche, Merck and Company; and serves in a leadership role for the American Association for Cancer Research, the International Association for the Study of Lung Cancer, the Society for Immunotherapy of Cancer and the Southwest Oncology Group. K.A.S. reports research funding from Takeda, Merck, Bristol Myers Squibb, AstraZeneca, Boehringer Ingelheim, Genentech/Roche, Akoya Biosciences and NextPoint Therapeutics; has received honoraria for consultant/advisory/speaker roles from Clinica Alemana de Santiago, AstraZeneca, EMD Serono, Takeda, Sanofi, Parthenon Therapeutics, Bristol Myers Squibb, Roche, Molecular Templates, Merck, Dynamicure, Indaptus, Moderna, Merus, PeerView, PER, Forefront Collaborative, DAINA and NextPoint Therapeutics. J.W. is currently employed by NEC Labs America. A.I.F. is currently employed by Caris. All other authors declare no competing interests.

## Additional information

**Extended data** is available for this paper at https://doi.org/10.1038/s41588-025-02351-7.

**Correspondence and requests for materials** should be addressed to Arutha Kulasinghe or David L. Rimm.

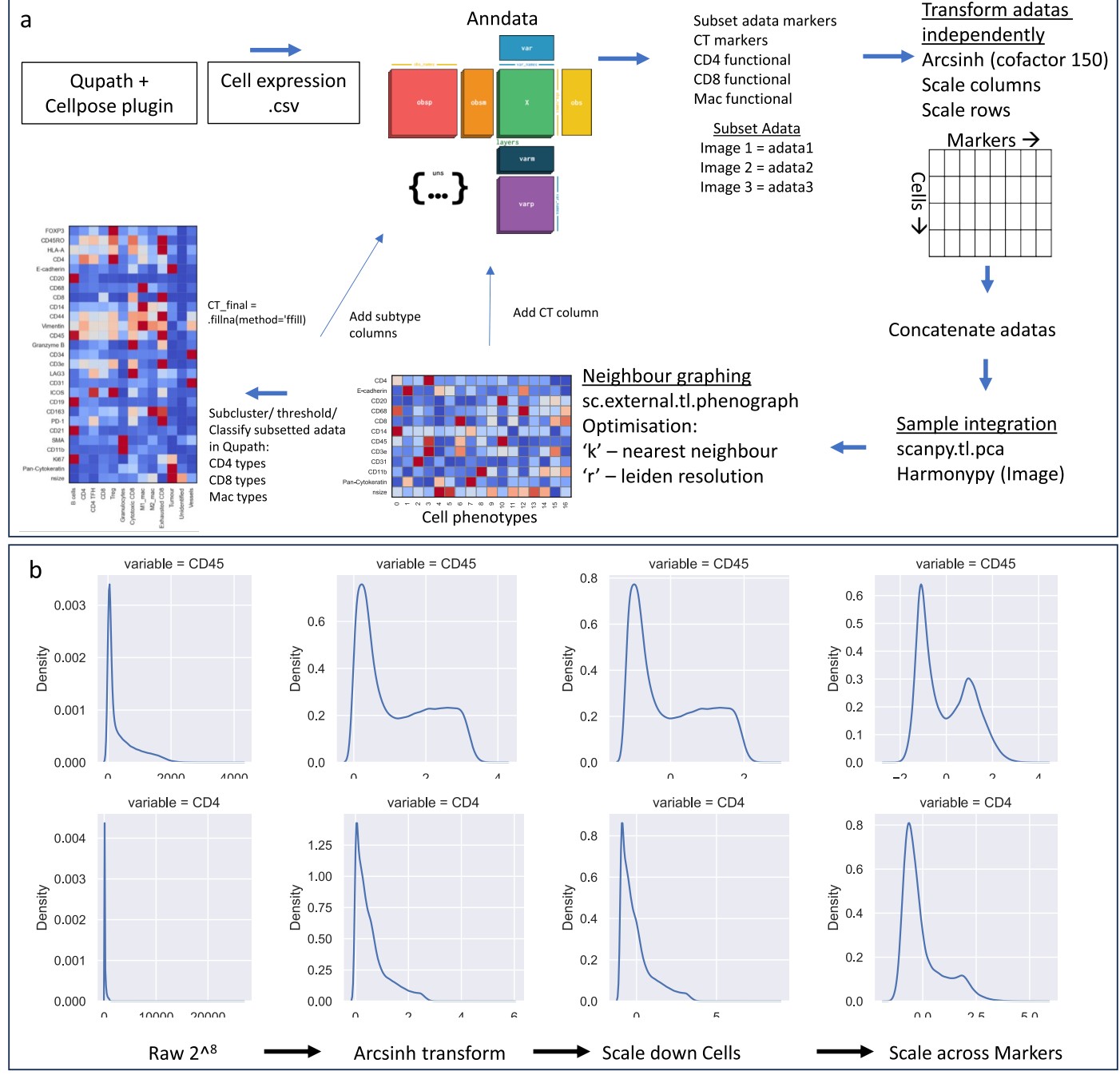

**Extended Data Fig. 1 | Workflow for CODEX image analysis and data preprocessing. (a)** The image workflow begins with exporting multiplex imaging data to QuPath, where cell segmentation is performed using the CellPose plugin. Resulting single-cell expression data is then exported as a .csv file and converted into an Anndata format. Selected markers are subsetted, and an arcsinh transformation (cofactor = 150) is applied independently to each dataset. Both row-wise (cells) and column-wise (markers) scaling are then conducted to standardize the data matrix (markers on the x-axis, cells on the y-axis). Multiple Anndata objects are concatenated across samples, and batch correction and image alignment are performed using Harmony after principal component analysis (PCA) via Scanpy. A nearest-neighbor graph is then constructed, and clustering is conducted using the sc.external.tl.phenograph function, tuning parameters such as the number of neighbors (K) and Leiden resolution (r), resulting in distinct cellular phenotypes. Cell type annotations are appended as metadata columns in the Anndata object. For specific populations such as CD4$^+$ T cells, CD8$^+$ T cells, and macrophages, sub-clustering and classification are further refined within QuPath. Final annotations are enriched using forward-fill ('ffill') methods to ensure completeness of subtype labels. **(b)** Raw fluorescence intensities (8-bit scale) are normalized using an arcsinh transformation to reduce skewness and handle the broad dynamic range. Expression values are subsequently scaled across both markers and cells. As shown using CD45 as an example, this preprocessing step prepares data for downstream analysis and comparison.

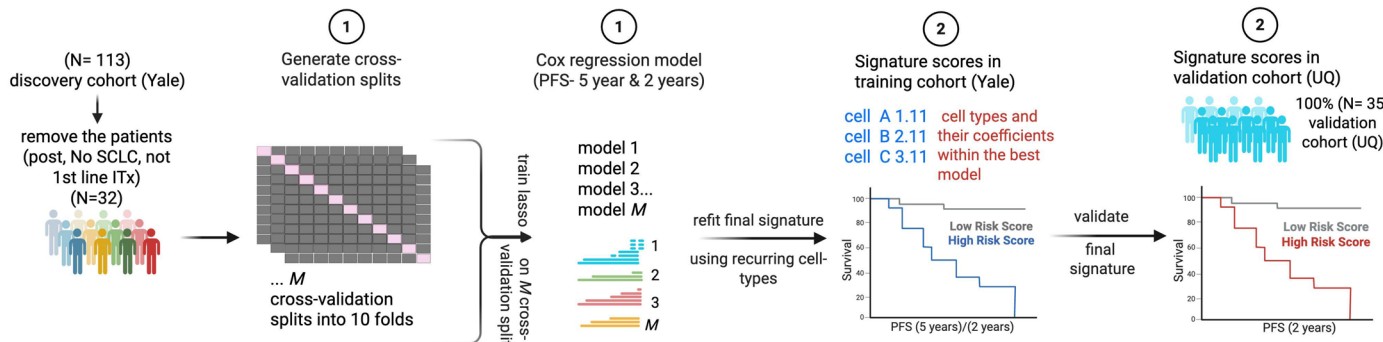

**Extended Data Fig. 2 | Schematic diagram for generating cell-type signatures associated with treatment outcome.** All signatures were trained using the Yale Cohort as our training cohort. To ensure robust model development, we generated M cross-validation splits and used these to train M models (1), each utilizing a LASSO Cox regression framework to predict 5-year progression-free survival (PFS). We set M = 50 for stroma and M = 100 for tumor, since we observed that these values were sufficient to identify a stable set of non-zero predictors for each compartment. Subsequently, we refit each signature using cell types with a non-zero coefficient in at least t of the models (2), where t was set to the maximum value such that each cell-type signature contained at least 3 cell types. We tested the final signatures on 100% of the training cohort to predict 2-year PFS (3). The validated model was then tested using the UQ validation cohort (4), thereby assessing its generalizability and predictive performance. The figure was created with BioRender.com.

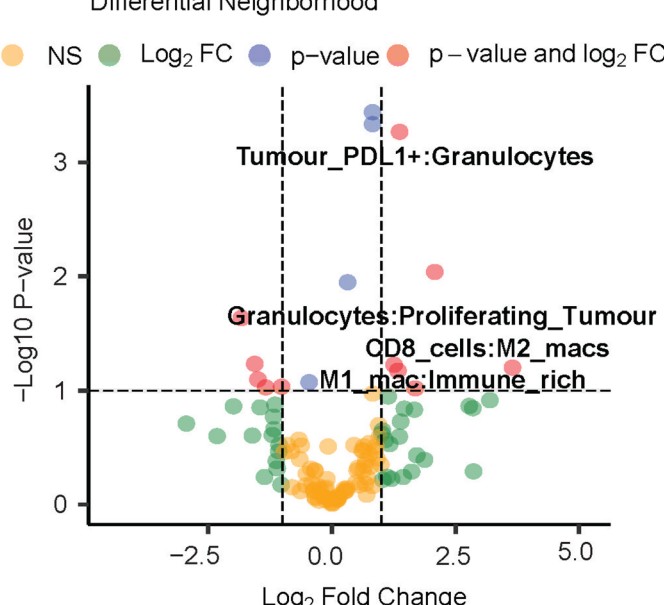

**Extended Data Fig. 3 | Neighborhood enrichment and correlations of PD-L1 expression with different cell types.** Differential cell type enrichment within each neighborhood is shown.

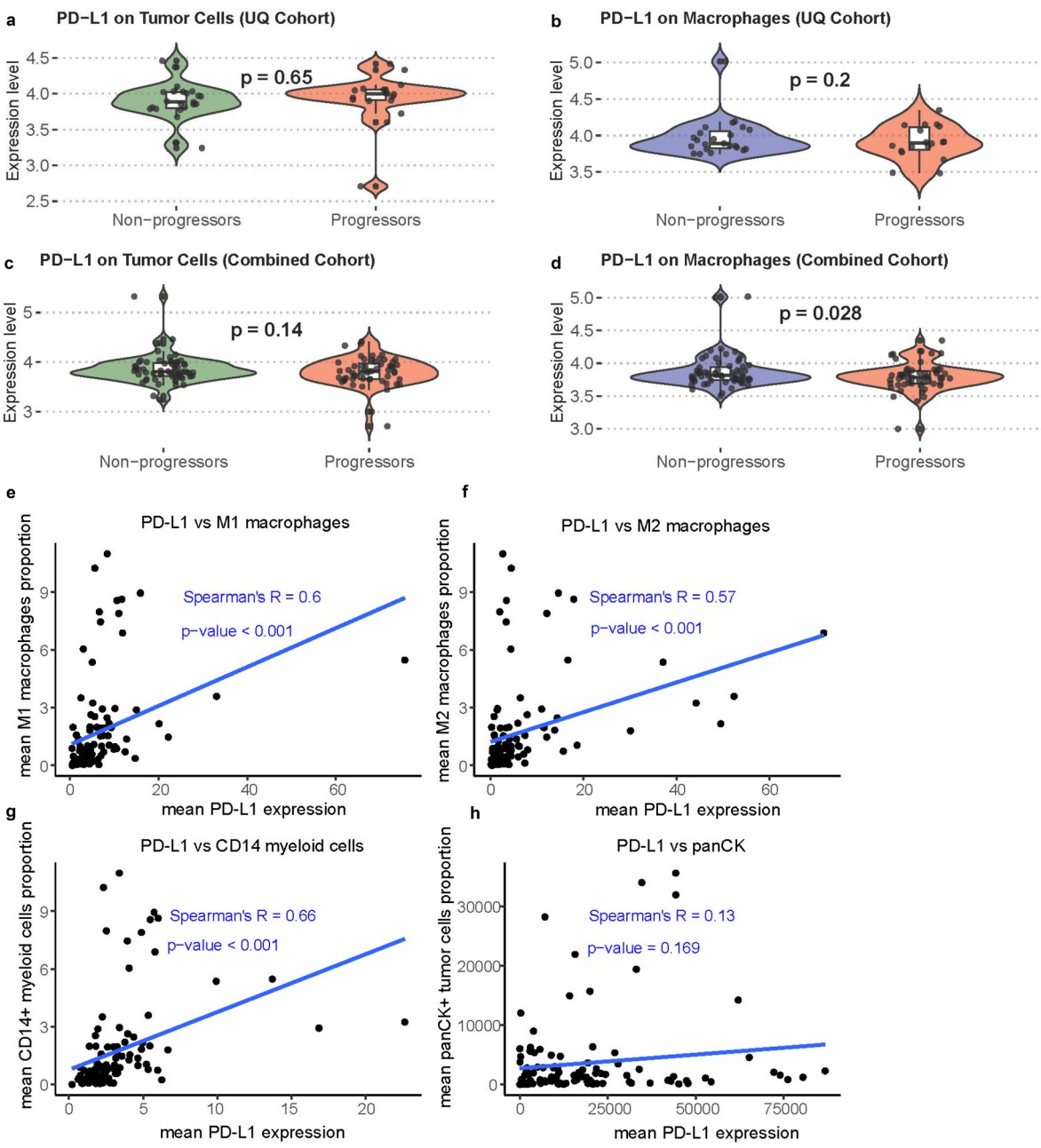

**Extended Data Fig. 4 | Comparison of PD-L1 expression on tumor cells and macrophages between progressors and non-progressors.** (**a**) PD-L1 expression on tumor cells in the UQ cohort, comparing progressors and non-progressors. (**b**) PD-L1 expression on macrophages in the UQ cohort. (**c**) PD-L1 expression on tumor cells in the combined Yale + UQ cohorts. (**d**) PD-L1 expression on macrophages in the combined Yale + UQ cohorts. One-tailed t-tests were employed in cases where the direction of the effect was hypothesized a priori. The correlation between PD-L1 expression and (**e**) M1 macrophages, (**f**) M2 macrophages, (**g**) CD14+ myeloid cells and (**h**) panCK+ tumor cells within the TIME, are shown, assessed using Spearman's rank correlation method.

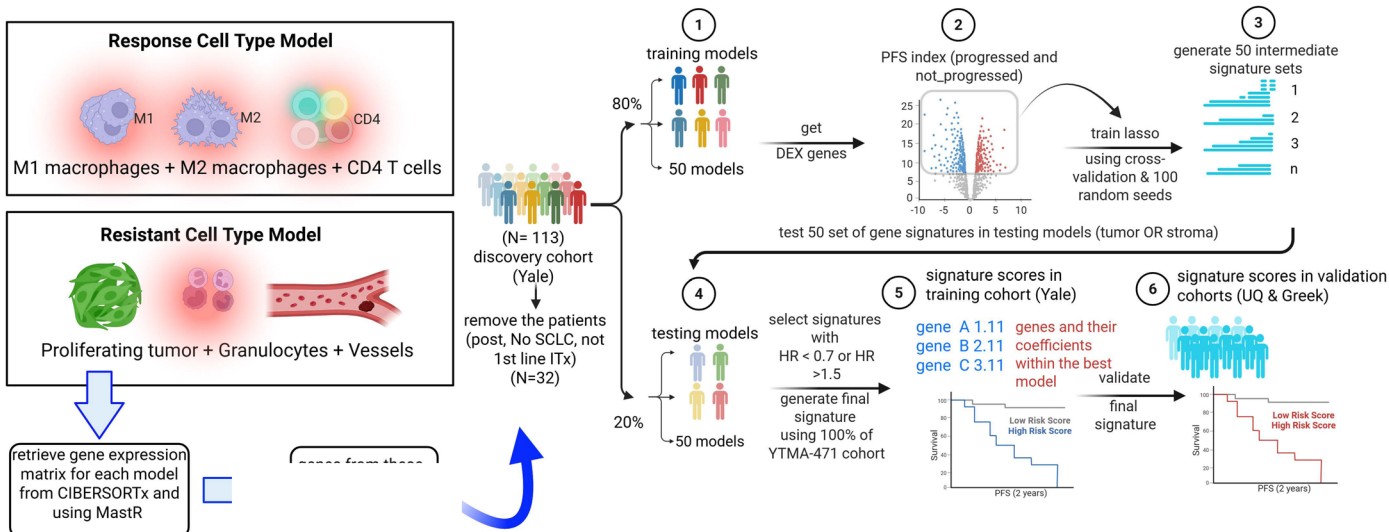

**Extended Data Fig. 5 | Schematic diagram for generating gene signatures from validated cell-type signatures.** All signatures were trained using the Yale cohort as our training cohort. We generated 50 splits of the cohort into training (80%) and validation (20%) subsets (1). To ensure robust model development, we generated intermediate signatures to predict 5-year progression-free survival (PFS) on each of these data splits: in each training split, (2) differentially expressed genes between progression-free and non-progression-free patients were identified, 100 LASSO Cox regression models were optimized with different random seeds, by first using 10-fold cross-validation on the current training split to select the penalty and then fitting a final LASSO Cox regression model with the best value of using the current training set, generating 50 intermediate signatures (3). The final signatures were then generated by selecting genes occurring in at least 0.05 of the models having HR > 1.5 (resistance) or HR < 0.7 (response) in their respective validation sets (4) and refitting the model using unpenalized Cox regression on the whole training set. Subsequently, we tested the consolidated model on 100% of the training cohort to predict 2-year PFS (5). The validated model was then tested using the UQ and Greek validation cohorts (6), thereby assessing its generalizability and predictive performance. The figure was created with BioRender.com.

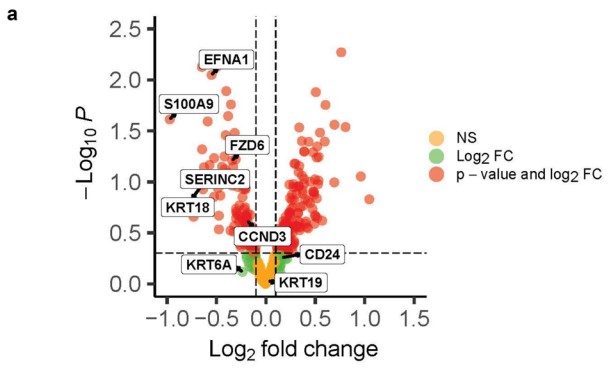

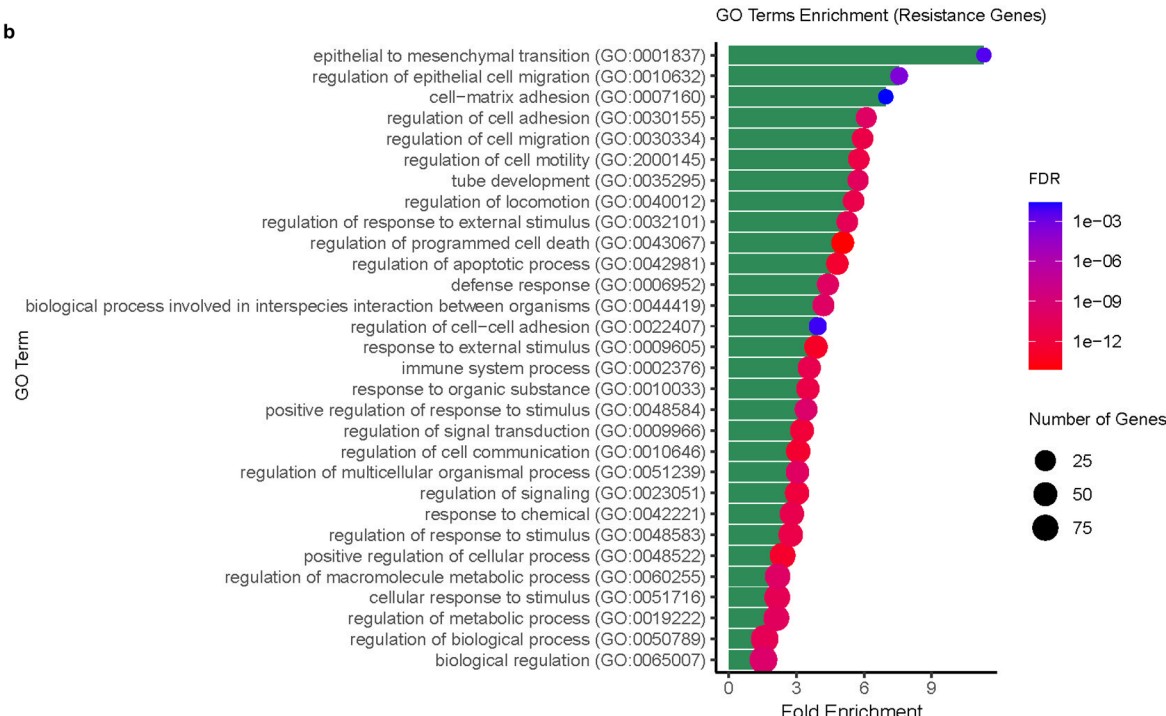

**Extended Data Fig. 6 | Differential gene expression analysis and Gene Ontology (GO) enrichment analysis for resistance signature.** (a) The volcano plot displays the differentially expressed genes derived from one of the 50 intermediate data splits. Each point represents a gene, with the x-axis indicating the log2 fold change and the y-axis showing the −log10 adjusted p-value. Genes within the final resistance signature are shown. Genes that are significantly differentially expressed are in red. The horizontal dashed line represents the significance threshold (adjusted p-value < 0.1), and the vertical dashed lines indicate the log2 fold change cutoffs. (b) The GO enrichment plot illustrates the functional categories significantly enriched among the differentially expressed genes. The y-axis represents the GO terms, and the x-axis shows the fold enrichment for each term. Enriched GO terms are biological processes. The size of the dots corresponds to the number of genes associated with each GO term, and the color gradient represents the significance level. These analyses highlight key pathways and processes potentially involved in the biological mechanisms underlying the observed gene expression changes, providing insights into the functional implications of the genes identified in the final resistance signature.

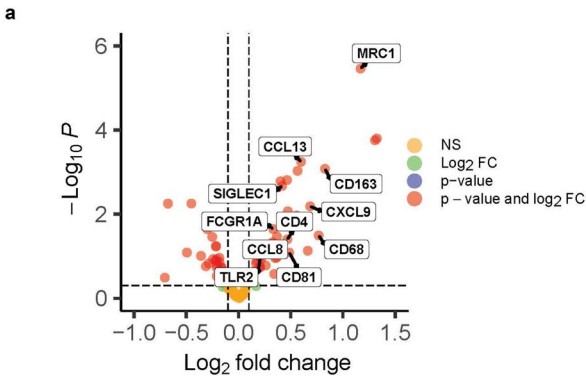

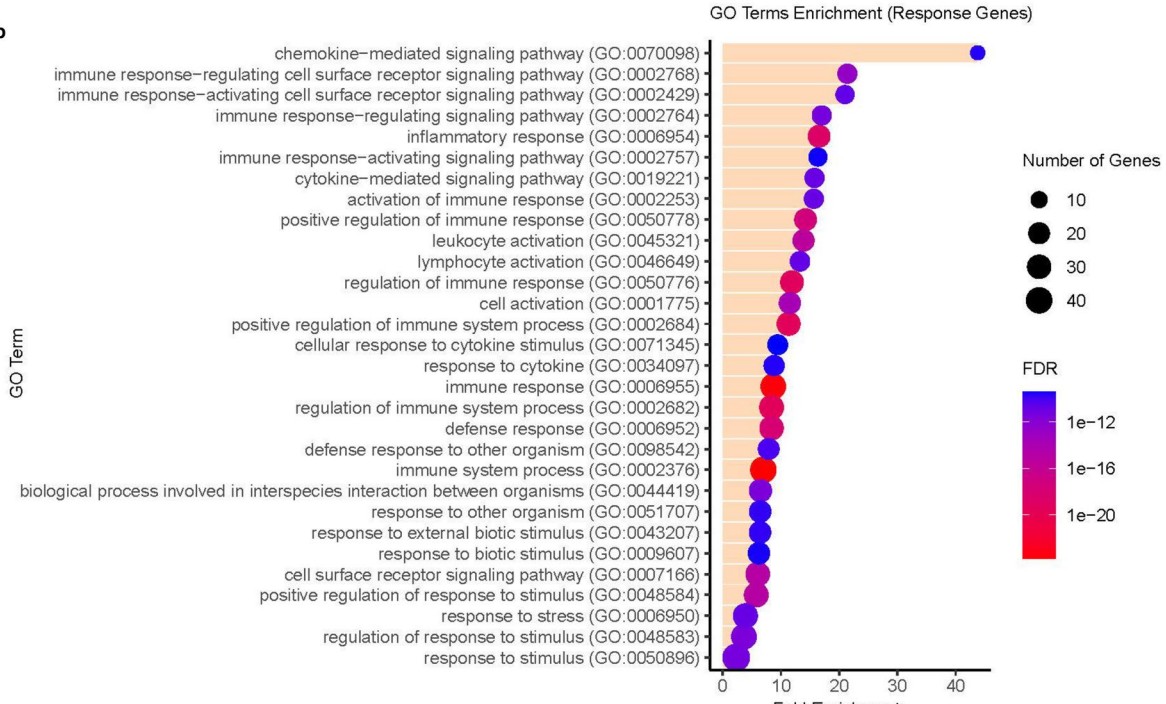

**Extended Data Fig. 7 | Differential gene expression analysis and Gene Ontology (GO) enrichment analysis for response signature.** (**a**) The volcano plot displays the differentially expressed genes derived from one of the 50 intermediate data splits. Each point represents a gene, with the x-axis indicating the log2 fold change and the y-axis showing the -log10 adjusted p-value. Genes within the final response signature are shown. Genes that are significantly differentially expressed are in red. The horizontal dashed line represents the significance threshold (adjusted p-value < 0.1), and the vertical dashed lines indicate the log2 fold change cutoffs. (**b**) The GO enrichment plot illustrates the functional categories significantly enriched among the differentially expressed genes. The y-axis represents the GO terms, and the x-axis shows the fold enrichment for each term. Enriched GO terms are biological processes. The size of the dots corresponds to the number of genes associated with each GO term, and the color gradient represents the significance level. These analyses highlight key pathways and processes potentially involved in the biological mechanisms underlying the observed gene expression changes, providing insights into the functional implications of the genes identified in the final response signature.

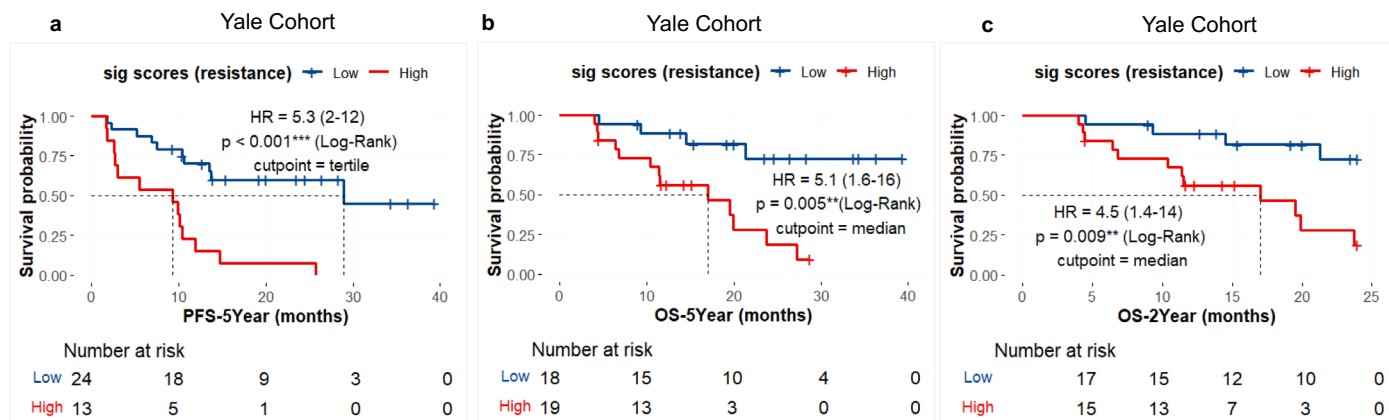

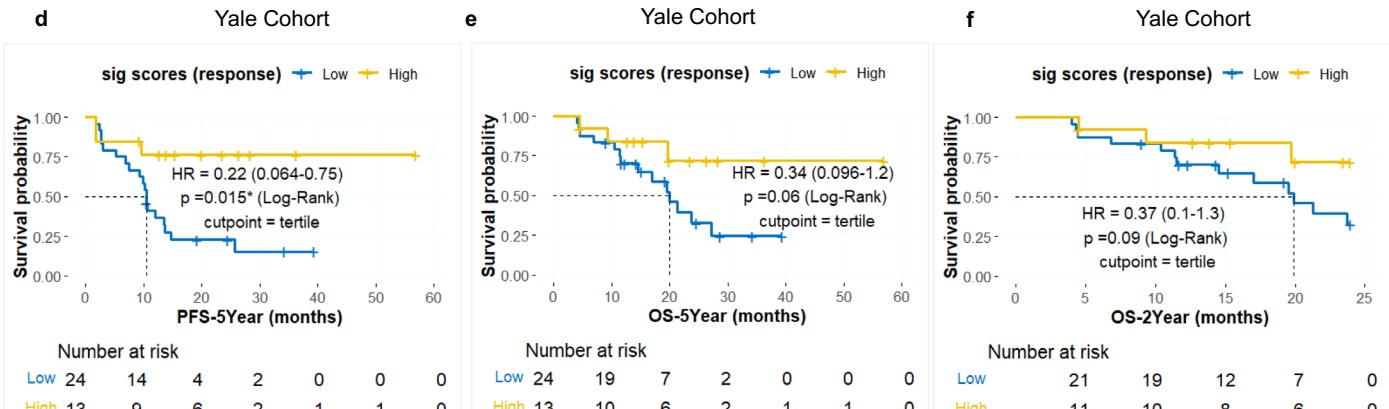

**Extended Data Fig. 8 | Testing of resistant and response gene signatures in tumor and stroma compartments of the Yale cohort to predict PFS-5 years, OS-5 years, and OS-2 years.** The performance of the resistant gene signature in the Yale cohort was evaluated for predicting (**a**) PFS-5 years, (**b**) OS-5 years and (**c**) OS-2 years in the tumor compartment, with KM survival curves illustrating its prognostic significance. Similarly, the response gene signature was tested in the stroma compartment of the Yale cohort for predicting (**d**) PFS-5 years, (**e**) OS-5 years and (**f**) OS-2 years, showing its predictive value for long-term and short-term survival outcomes. Two-tailed and one-tailed log-rank tests are used on the discovery and validation cohorts, respectively, where the direction of the effect in the latter is chosen to match the direction of the effect in the former. Two-tailed log-rank tests are used.

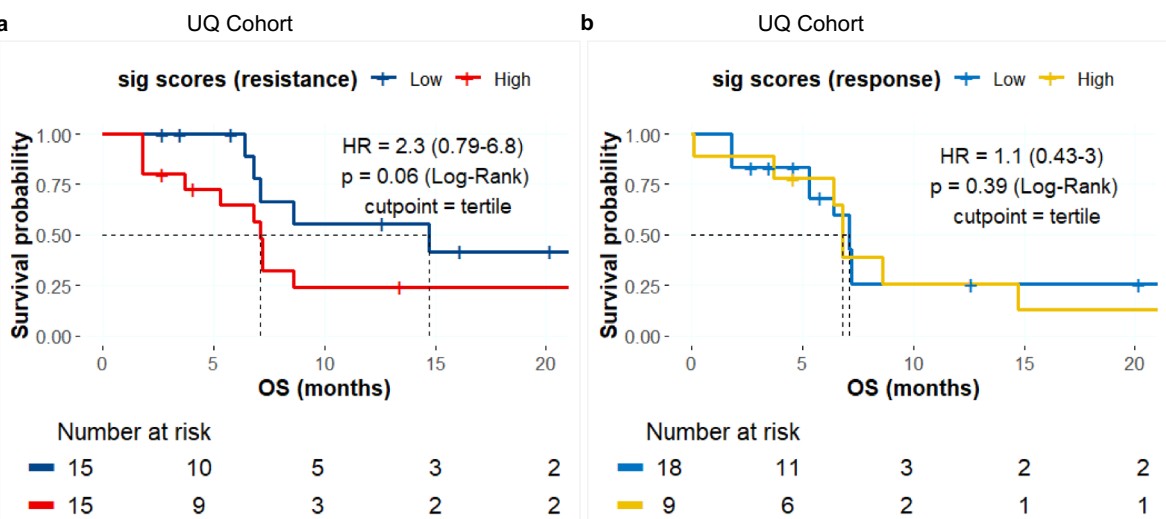

**Extended Data Fig. 9 | Validation of resistant and response gene signatures in tumor and stroma compartments of the UQ cohort to predict overall survival.** (**a**) The resistant gene signature was validated in the tumor compartment of the UQ cohort to predict OS. The KM survival curve demonstrates the performance of the signature for stratifying patients based on survival outcomes. (**b**) The response gene signature was validated in the stroma compartment of the UQ cohort to predict OS. The KM survival curve shows the performance of the signature. Two-tailed and one-tailed log-rank tests are used on the discovery and validation cohorts, respectively, where the direction of the effect in the latter is chosen to match the direction of the effect in the former.

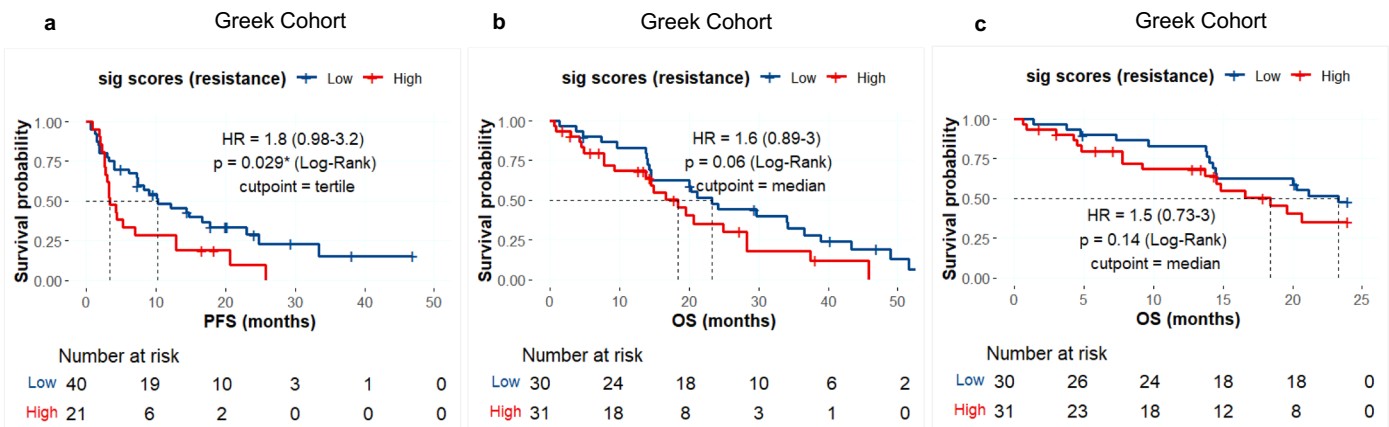

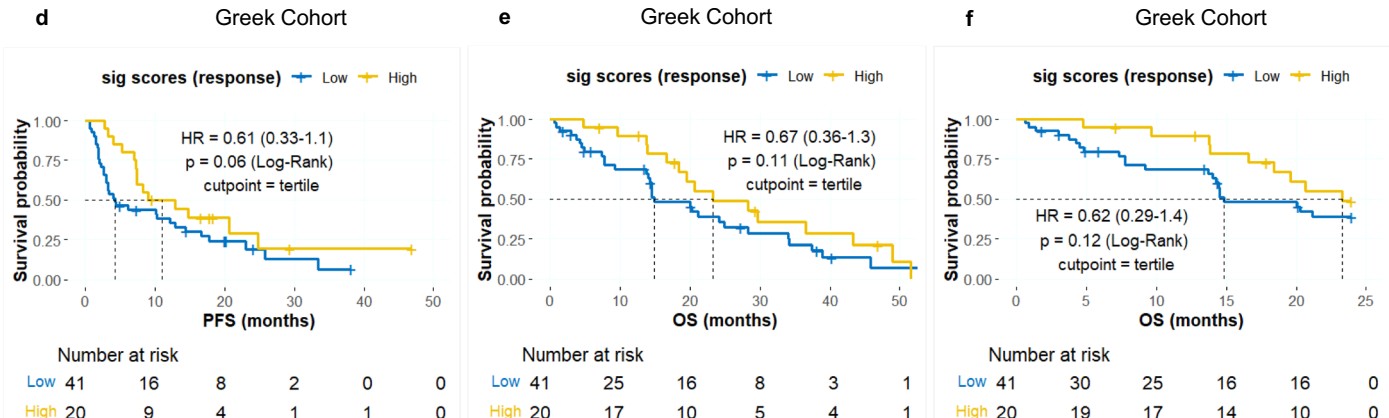

**Extended Data Fig. 10 | Validation of resistant and response gene signatures in tumor and stroma compartments of the Greek cohort to predict PFS (5 years) and OS (5 years and 2 years).** The performance of the resistant gene signature in the Greek cohort was evaluated for predicting (**a**) PFS-5 years, (**b**) OS-5 years and (**c**) OS-2 years in the tumor compartment, with KM survival curves illustrating its prognostic significance. Similarly, the response gene signature was tested in the stroma compartment of the Greek cohort for predicting (**d**) PFS-5 years, (**e**) OS-5 years and (**f**) OS-2 years, showing its predictive value for long-term and short-term survival outcomes. Two-tailed and one-tailed log-rank tests are used on the discovery and validation cohorts, respectively, where the direction of the effect in the latter is chosen to match the direction of the effect in the former.

# Reporting Summary

## Statistics

For all statistical analyses, confirm that the following items are present in the figure legend, table legend, main text, or Methods section.

| n/a | Confirmed | |
|---|---|---|
| ☐ | ☒ | The exact sample size (*n*) for each experimental group/condition, given as a discrete number and unit of measurement |
| ☐ | ☒ | A statement on whether measurements were taken from distinct samples or whether the same sample was measured repeatedly |
| ☐ | ☒ | The statistical test(s) used AND whether they are one- or two-sided *Only common tests should be described solely by name; describe more complex techniques in the Methods section.* |
| ☐ | ☒ | A description of all covariates tested |
| ☐ | ☒ | A description of any assumptions or corrections, such as tests of normality and adjustment for multiple comparisons |
| ☐ | ☒ | A full description of the statistical parameters including central tendency (e.g. means) or other basic estimates (e.g. regression coefficient) AND variation (e.g. standard deviation) or associated estimates of uncertainty (e.g. confidence intervals) |
| ☐ | ☒ | For null hypothesis testing, the test statistic (e.g. *F*, *t*, *r*) with confidence intervals, effect sizes, degrees of freedom and *P* value noted *Give P values as exact values whenever suitable.* |
| ☒ | ☐ | For Bayesian analysis, information on the choice of priors and Markov chain Monte Carlo settings |
| ☐ | ☒ | For hierarchical and complex designs, identification of the appropriate level for tests and full reporting of outcomes |
| ☒ | ☐ | Estimates of effect sizes (e.g. Cohen's *d*, Pearson's *r*), indicating how they were calculated |

*Our web collection on statistics for biologists contains articles on many of the points above.*

## Software and code

Policy information about availability of computer code

| Data collection | Commercial Software<br>1) Akoya Biosciences PCF (Phenocycler Fusion) Platform: This platform's associated software was used for the analysis of spatial proteomic data, including image processing and data export in qptiff format for further analysis.<br>2) GeoMx DSP (Digital Spatial Profiling) Platform by Nanostring Technologies: The software provided by Nanostring was used for the analysis of whole transcriptome (WTA) and cancer transcriptome (CTA) data, including processing sequencing data and generating digital count conversion (.dcc) files. |
|---|---|
| Data analysis | Open-Source Software<br>1) QuPath (v0.4.2), 2) Cellpose (v2.0), 3) Scanpy (Version: 1.8.2), 4) Harmonypy (v 0.0.5), 5) Phenograph (v1.5.7), 6) CIBERSORTx, 7) Scimap, 8) R (v4.2.1), 9) mastR (v1.8.0), and 10) glmnet (v4.1-9).<br>Custom Scripts<br>Custom scripts were developed for data preprocessing, integration, clustering, and the development of cell type and gene signatures. These scripts include the processing pipelines for the Phenocycler Fusion data, as well as the Cox regression modeling with LASSO regularization. These scripts are available on GitHub at https://github.com/tznaung/NSCLC_SpatialOmics. |

For manuscripts utilizing custom algorithms or software that are central to the research but not yet described in published literature, software must be made available to editors and reviewers. We strongly encourage code deposition in a community repository (e.g. GitHub). See the Nature Portfolio guidelines for submitting code & software for further information.

# Data

Policy information about availability of data

All manuscripts must include a data availability statement. This statement should provide the following information, where applicable:

- Accession codes, unique identifiers, or web links for publicly available datasets
- A description of any restrictions on data availability
- For clinical datasets or third party data, please ensure that the statement adheres to our policy

Raw and processed DSP-GeoMx WTA RNA sequencing data from the Yale discovery cohort and the Greek validation cohort are available under accession number GSE271689. DSP-CTA raw RNA sequencing data of UQ validation cohort can be assessed via GSE221733. Further data requests should be directed to the corresponding authors.

# Research involving human participants, their data, or biological material

Policy information about studies with human participants or human data. See also policy information about sex, gender (identity/presentation), and sexual orientation and race, ethnicity and racism.

| | |
|---|---|
| Reporting on sex and gender | Sex-Based Analysis<br>The study did not specifically report sex-based analyses due to the primary focus on developing and validating cell type and gene signatures associated with immunotherapy outcomes. The datasets used for this analysis were primarily stratified by clinical outcomes (e.g., progression-free survival) rather than sex or gender. Therefore, the influence of sex or gender on these outcomes was not assessed in the primary analyses.<br>Justification for Lack of Sex- and Gender-Based Analysis<br>The absence of sex- and gender-based analyses in this study is due to the research design, which aimed to develop predictive models based on cell type and gene expression profiles across the entire patient population. Given the exploratory nature of this study and the focus on biomarker discovery, the analysis was not stratified by sex or gender. Future studies may consider sex- and gender-specific differences if they are deemed relevant to the specific biomarkers or treatment responses being investigated. |
| Reporting on race, ethnicity, or other socially relevant groupings | Race, ethnicity, and other socially relevant groupings were not explicitly analyzed in this study. To control for confounding variables, we focused on patients receiving PD-1 based immunotherapy as their first-line treatment, standardizing treatment exposure across cohorts. We also used machine learning models and Cox regression with LASSO regularization, which inherently controls for multiple variables simultaneously, reducing bias. Additionally, cell type and gene signature analyses were performed independently of demographic factors, aiming to identify robust biomarkers associated with treatment outcomes, regardless of race or ethnicity. Future studies may explore these factors where relevant. |
| Population characteristics | Population Characteristics<br><br>The study included three independent cohorts of NSCLC patients treated with PD-1 based immunotherapies.<br>Yale Cohort (Training Set)<br>Sample Size: 113 tissue samples, narrowed to 41 first-line only immunotherapy treated patients for final analysis<br>Age: Median age of participants was 69.<br>Diagnosis: Advanced or metastatic NSCLC.<br>Treatment: First-line PD-1 based immunotherapy.<br>Covariates: Included clinical characteristics such as age, sex, tumor stage, histology, prior chemotherapy, smoking status, type of immunotherapy and treatment history.<br><br>UQ Cohort (Validation Set)<br>Sample Size: 42 tissue samples, narrowed to patients for final analysis.<br>Age: Median age of participants was 63.<br>Diagnosis: Recurrent NSCLC post-surgery.<br>Treatment: First-line PD-1 based immunotherapy.<br><br>Greek Cohort Cohort (Training Set)<br>Sample Size: 79 tissue samples, narrowed to 61 first-line only immunotherapy treated patients for final analysis.<br>Age: Median age of participants was 70.<br>Diagnosis: Advanced or metastatic NSCLC.<br>Treatment: First-line PD-1 based immunotherapy.<br>Covariates: Included clinical characteristics such as age, sex, tumor stage, histology, prior chemotherapy, smoking status, type of immunotherapy and treatment history. |
| Recruitment | Tissue samples were retrospectively collected from Yale Cancer Center (YCC), the University of Queensland (UQ), and the University of Anthems (Greece). The Yale cohort included samples collected between 2012 and 2019, the UQ cohort between 2009 and 2018, and the Greek cohort between 2019 and 2023. Retrospective collection was employed to minimize prospective bias and ensure representative data aligned with the study objectives. |
| Ethics oversight | The study protocol was approved by the Yale Human Investigation Committee (Protocol #95050082199) for the Yale Cancer Center cohort; by the Queensland University of Technology Human Research Ethics Committee (Protocol #2000000494), ratified by the University of Queensland, for the UQ cohort; and by the Ethics Committee of Sotiria General Hospital, Medical School (HIC Protocol #16760/23-06-2023) for the Greek cohort. |

Note that full information on the approval of the study protocol must also be provided in the manuscript.

# Field-specific reporting

Please select the one below that is the best fit for your research. If you are not sure, read the appropriate sections before making your selection.

☒ Life sciences ☐ Behavioural & social sciences ☐ Ecological, evolutionary & environmental sciences

For a reference copy of the document with all sections, see nature.com/documents/nr-reporting-summary-flat.pdf

# Life sciences study design

All studies must disclose on these points even when the disclosure is negative.

| Sample size | No statistical method was used to predetermine sample size. Sample sizes were based on the availability of tissue samples collected during the specified periods: 2012–2019 for the Yale cohort, 2009–2018 for the University of Queensland cohort, and 2019–2023 for the Greek cohort. All available samples that met the inclusion criteria were included in the analysis to maximize the use of existing retrospective data and resources. |
|---|---|
| Data exclusions | One region of interest (ROI) was excluded from the analyses due to data inconsistencies observed between two operators (TNA and MM) during the analysis of gene expression patterns using the DSP-GeoMx-WTA platform. Although potential issues were identified prior to sequencing, the experiment was carried out to determine whether the data would remain usable. Results from this ROI differed significantly from the rest of the dataset and were deemed unreliable for inclusion. This exclusion was predetermined and is documented in Extended Data Figure. 8. The remaining data were not excluded and were used to develop gene signatures of resistance and response using a LASSO-based framework within the tumor compartment of advanced NSCLC. |
| Replication | To verify the reproducibility of the experimental findings from the Yale NSCLC cohort, we validated the LASSO-generated cell type models in an external cohort from the University of Queensland (UQ), Australia. All replication attempts were successful, confirming the generalizability of the identified cell-type signatures. For the gene signatures, validation was performed across two independent external cohorts from Australia and Europe. These replication efforts were also successful, supporting the robustness, reproducibility, and broader applicability of the identified gene models. |
| Randomization | In this study, participants were not prospectively randomized into experimental groups. Instead, during model development, samples from the Yale cohort were randomly partitioned into training and testing sets as part of a cross-validation strategy within the LASSO framework. This re-sampling approach helped to minimize bias and control for covariates during the development of resistance and response-specific cell type and gene signatures. Given the retrospective nature of the study, this method provided a rigorous and reproducible means of model evaluation. |
| Blinding | Blinding was not applicable to this study, as the focus was on analyzing pre-existing tissue samples and gene expression data using computational methods. Since the study involved retrospective data analysis rather than intervention-based experimentation, group allocation and investigator bias were minimized by the automated, objective nature of the LASSO framework. |

# Reporting for specific materials, systems and methods

We require information from authors about some types of materials, experimental systems and methods used in many studies. Here, indicate whether each material, system or method listed is relevant to your study. If you are not sure if a list item applies to your research, read the appropriate section before selecting a response.

### Materials & experimental systems

| n/a | Involved in the study |
|---|---|
| ☐ | ☒ Antibodies |
| ☒ | ☐ Eukaryotic cell lines |
| ☒ | ☐ Palaeontology and archaeology |
| ☒ | ☐ Animals and other organisms |
| ☐ | ☒ Clinical data |
| ☒ | ☐ Dual use research of concern |
| ☒ | ☐ Plants |

### Methods

| n/a | Involved in the study |
|---|---|
| ☒ | ☐ ChIP-seq |
| ☒ | ☐ Flow cytometry |
| ☒ | ☐ MRI-based neuroimaging |

# Antibodies

| Antibodies used | In this study, the following nanostring antibodies were used to define regions of interest (ROIs) by fluorescent masking.<br>1. CD45 (leukocyte marker): Supplier NanoString Technologies<br>2. PanCK (Tumor compartment marker):Supplier: NanoString Technologies<br>3. CD68 (Macrophage marker):Supplier: NanoString Technologies<br>The following Akoya antibodies were used for for cell phenotyping, functional subset identification by CODEX<br>4. CD31 (Endothelial cell marker):Supplier: Akoya Biosciences |
|---|---|

5. CD4 (T helper cell marker):Supplier: Akoya Biosciences
6. HLA-A (human leukocyte antigen): Supplier: Akoya Biosciences
7. CD44 (cancer stem cells): Supplier: Akoya Biosciences
8. CD20 (B-cell marker):Supplier: Akoya Biosciences
9. SMA (Smooth Muscle Actin marker):Supplier: Akoya Biosciences
10. E-cadherin (Epithelial cell marker):Supplier: Akoya Biosciences
11. CD68 (Macrophage marker):Supplier: Santa Cruz; clone; KP1, catalog: sc-20060
12. CD45RO (memory T cells marker): Supplier: Akoya Biosciences
13. Pan-cytokeratin (epithelial marker):Supplier:Akoya Bioscience
14. CD45 (leukocyte marker): Supplier Akoya Biosicences
15. Vimentin (Mesenchymal cell marker):Supplier: Akoya Biosciences
16. CD11b (Myeloid cell marker):Supplier: Akoya Biosciences
17. CD11c (Dendritic cell marker):Supplier: Akoya Biosciences
18. CD34 (Endothelial progenitor cell marker):Supplier: Akoya Biosciences
19. CD8 (Cytotoxic T-cell marker):Supplier: Akoya Biosciences
20. IDO1(Tryptophan regulation):Supplier:Akoya Biosciences
21. FOXP3 (T regulatory cell marker):Supplier: Akoya Biosciences
22. CD21 (B cells and follicular dendritic cells marker): Supplier: Akoya Biosciences
23. CD14 (Monocyte/Macrophage marker):Supplier: Akoya Biosciences
24. Granzyme B (Cytotoxic T-cell marker):Supplier: Akoya Biosciences
25. Collagen IV (an antibody that binds to Type IV collagen): Supplier: Akoya Biosciences
26. PD-L1 (Tumor cell marker):Supplier: Akoya Biosciences
27. CD3e (T-cell marker):Supplier: Akoya Biosciences4
28. PD-1 (Exhausted T-cell marker):Supplier: Akoya Biosciences
29. Ki67 (Proliferation marker):Supplier: Akoya Biosciences
30. ICOS (T Follicular helper cell marker):Supplier: Akoya Biosciences
31. LAG3 (immune checkpoint): Supplier: Akoya Biosciences
32. CD163 (M2 Macrophage marker):Supplier: Akoya Biosciences

Validation

The validation of each primary antibody used in this study was conducted as follows:
Antibodies from Akoya Biosciences and NanoString Technologies: All antibodies obtained from Akoya Biosciences and NanoString Technologies were fully validated by the manufacturers for the species and applications used in this study. These antibodies are routinely employed in their respective platforms (Phenocycler Fusion and GeoMx DSP) and have been rigorously tested to ensure specificity and reliability. Validation statements provided by the manufacturers affirm their appropriateness for the species and experimental conditions applied in our research. Application note for antibody validation by nanostring can be found: https://nanostring.com/wp-content/uploads/WP_GeoMx_Antibody_Validation_White_Paper.pdf.
application note for antibody validation for akoya can be found https://www.akoyabio.com/wp-content/uploads/2022/01/Phenocycler_Technical-Note_Validation-of-Commercial_DN-00140.pdf aswell as in original CODEX Cell publication https://pubmed.ncbi.nlm.nih.gov/32763154/
CD68 (Clone KP1) from Santa Cruz Biotechnology: The CD68 antibody (Clone KP1) sourced from Santa Cruz Biotechnology has been validated by the manufacturer for human tissue and immunohistochemical applications. This antibody is widely recognized in the scientific community and has been cited in over 200 peer-reviewed publications, demonstrating its reliability and effectiveness in similar studies. The extensive use and validation of this clone across numerous studies further support its suitability for the applications in our research.

# Clinical data

Policy information about clinical studies

All manuscripts should comply with the ICMJE guidelines for publication of clinical research and a completed CONSORT checklist must be included with all submissions.

Clinical trial registration

This study is retrospective and does not involve a clinical trial; therefore, it does not have a trial registration number.

Study protocol

As this study is retrospective and not a clinical trial, a formal trial protocol is not available.

Data collection

Data collection took place in three primary settings:

1. Yale Cancer Center (YCC), New Haven, CT, USA
Setting: A comprehensive cancer center within a large academic medical institution.
Time Period: Tissue samples were collected retrospectively between 2012 and 2019.

2. University of Queensland (UQ), Brisbane, Australia
Setting: A leading research university and affiliated hospitals specializing in oncology.
Time Period: Tissue samples were collected retrospectively between 2009 and 2018.

3. Sotiria General Hospital, Athens, Greece
Setting: A major academic hospital affiliated with the Medical School in Greece, specializing in pulmonary diseases and oncology.
Time Period: Tissue samples were collected retrospectively between 2019 and 2023.

Outcomes

Primary Outcome Measures
The primary outcome measure was the progression-free survival (PFS) at 2 years in patients with non-small cell lung cancer (NSCLC) treated with PD-1 based immunotherapies. PFS was defined as the time from the start of treatment to disease progression or death from any cause. This measure was assessed using clinical data and imaging studies in accordance with the Response Evaluation Criteria In Solid Tumors (RECIST) version 1.1.

Secondary Outcome Measures
The secondary outcome measures included the development and validation of cell type-specific and gene expression signatures predictive of treatment response or resistance. These measures were assessed using spatial proteomic and transcriptomic profiling data, processed and analyzed through a LASSO-based Cox regression framework. The performance of these signatures was evaluated by their ability to predict PFS in independent validation cohorts.
Both outcome measures were pre-defined based on the study's objectives to understand the biomarkers associated with immunotherapy outcomes in NSCLC.

## Plants

| | |
|---|---|
| Seed stocks | not applicable |
| Novel plant genotypes | not applicable |
| Authentication | not applicable |

