## [Peer Review File · Nature Genetics]

Spatial Signatures for Predicting Immunotherapy Outcomes using Multi-Omics in Non-Small Cell Lung Cancer

Corresponding Author: Dr David Rimm

A version of this paper was originally rejected for publication by Nature Genetics, however that decision was reconsidered after appeal by the authors.

Version 0:

Decision Letter:

15th Oct 2024

Dear David,

Your Article entitled "Enhancing Immunotherapy Outcomes: Spatial Multi-Omics Models for Non-Small Cell Lung Cancer" has now been seen by 2 referees, whose comments are attached. While they find your work of potential interest, they have raised serious concerns which in our view are sufficiently important that they preclude publication of the work in Nature Genetics, at least in its present form.

While the referees find your work of some interest, they raise concerns about the strength of the novel conclusions that can be drawn at this stage.

Reviewer #1 says this is a well-written and performed study but has a number of areas for improvement. They note the heterogeneity of the small cohorts, as well as the low predictive power of your models.

Referee #2 acknowledges the potential of your work but straightforwardly thinks that the substantial cohort heterogeneity makes it impossible to judge whether any of your findings will be generalizable.

We note that both of these reports highlight the cohort heterogeneity and model generalisability, and we agree that these are major concerns - and we are unwilling to overrule the referees on such a fundamental issue. In our estimation, it seems hard to see how this limitation could be easily addressed; but given there is some enthusiasm voiced from the reviewers, we would be interested in a revision that is able to show your findings do apply beyond these specific data.

Should further experimental data allow you to fully address these criticisms we would be willing to consider an appeal of our decision (unless, of course, something similar has by then been accepted at Nature Genetics or appeared elsewhere). This includes submission or publication of a portion of this work someplace else.

The required new experiments and data include, but are not limited to those detailed here. We hope you understand that until we have read the revised manuscript in its entirety we cannot promise that it will be sent back for peer review.

If you are interested in attempting to revise this manuscript for submission to Nature Genetics in the future, please contact me to discuss a potential appeal. Otherwise, we hope that you find our referees' comments helpful when preparing your manuscript for resubmission elsewhere.

Sincerely,

Michael Fletcher, PhD
Senior Editor, Nature Genetics
ORCID: 0000-0003-1589-7087

Referee expertise: lung cancer, immunotherapy, spatial -omics.

Reviewers' Comments:

Reviewer #1 (Remarks to the Author):

In this manuscript, Aung, Monkman, Warrell and colleagues used a multi-omics approach leveraging spatial proteomics (CODEX) and transcriptomics (DSP-GeoMx-WTA) to profile the tumor immune microenvironment in two independent, pre-PD-1 non-small cell lung cancer datasets. Spatial proteomic analysis identified a resistance cell type model, characterized by proliferating tumor cells, granulocytes, and vessels. By contrast, a response cell type model was characterized by higher levels of M1 and M2 macrophages, as well as CD4 T cells. GeoMx was then used to derive gene signatures from these cell type models (e.g., resistance model – EMT and cell migration genes; response model – immunomodulatory genes).

Overall, this is a well-written and performed study. I had the following comments/questions:

1. The introduction emphasizes the importance of focusing on 1st line cohorts to minimize confounding from previous treatment regimens; however, the Queensland ICI cohort appears to include both 1st and 2nd line patients.
2. The Queensland cohort also consists of surgically resected specimens from patients who subsequently recurred. What steps were taken to ensure that these were recurrences versus new primary tumors (and thus that the specimens used in this analysis are representative of the pre-PD-1 tumors?)
3. While the manuscript and supplemental Table 1 references 113 patients in the Yale cohort, the actual number of patients with adequate tissue who met the eligibility criteria for inclusion was only 37-41. The second cohort was only 31 to 35 patients. Overall, these are relatively small sample sizes. It would also be advisable to provide the clinical characteristics of only those patients who were actually included in Table S1.
4. Supplemental Table 1 – In the Queensland group, 50% were treated in the 2nd line, but the table lists only 11.9% treated in “2nd-line or beyond.” Presumably this should be “beyond 2nd-line”?
5. One major question is why the authors used 2-year PFS as their primary endpoint as opposed to objective response rate? Figure 2E-H treats this as a binary endpoint but obviously some patients will have much more durable disease control than others. Likewise, at various points in the manuscript, the authors use the nomenclature of “responders” versus “non-responders” but it appears that they are referring to those who are progression-free versus those who are not progression-free.
6. Proliferating tumor doesn't appear in Figure 2B. How was this then defined? Ki67? Other features?
7. One of the core features of the resistance cell type model is proliferating tumor cells. This is somewhat intuitive in that tumors with more highly proliferative tumor cells would presumably do worse. A central question is whether this is simply a prognostic factor and unrelated to IO outcomes? I would add that the KM curves in Figure 3F, while significant, are underwhelming, and show the curves crossing.
8. The validation of the response cell type model is also underwhelming, as the KMs appear to cross. On a related point, can the authors provide context for using a 1-sided log-rank for the UQ cohort and 2-sided for the Yale cohort?
9. Page 26, lines 4654-466: “The lack of statistical significance in OS further emphasizes the specificity of the response signature for predicting disease control and progression.” This is a rather optimistic view. An alternative explanation is that the model has modest predictive abilities that don't carry over to OS.
10. M2 and PD-L1+ appear to be in opposite directions of what has been seen clinically.
11. Page 29, lines 518, “These findings suggest that the presence of PD-L1 on M1 and M2 macrophages, rather than on tumor cells, could be indicative of a positive response to treatment.” While the discussion section on the benefits of CPS versus TPS are certainly reasonable, I would caution the authors from over-interpreting their data given the very small sample size – especially since they didn't show that tumor cell PD-L1 positive correlated with outcomes (but it has in many large phase III clinical trials).
12. Figure 5c – There is a fair bit of overlap in M2 fractions between those with progression and those without progression.
13. In the limitations section, it's the first time we see it mentioned that some patients received chemo-immunotherapy but it's unclear how many patients? Supplemental Table 1 doesn't list chemo-immunotherapy.

Reviewer #2 (Remarks to the Author):

Enhancing Immunotherapy Outcomes: Spatial Multi-Omics Models for Non-Small Cell Lung Cancer by Aung and Monkman, et al. presents an innovative approach to predicting patient responses to immunotherapy in non-small cell lung cancer. By integrating spatial proteomic and transcriptomic profiling, the study aims to identify specific cell types and gene signatures within the tumor immune microenvironment associated with treatment resistance and response. The study's strengths include its multi-omics approach and the application of machine learning, both of which have the potential to enhance precision oncology.

The major significant limitation of this study is the heterogeneity of the dataset. The cohort includes patients with varying treatment regimens, disease stages, and other clinical variables such as age, gender, ethnicity, smoking history, performance status, comorbidities, tumor stage, and prior treatments. Additionally, differences in immunotherapy types,

dosages, and concomitant treatments further contribute to the variability. This introduces numerous confounding variables that complicate the interpretation of the results. Without adequate control for these factors—such as through multivariable regression analysis or stratification—the associations between immune microenvironment features and clinical outcomes may be confounded by unaccounted influences. Consequently, the predictive value of the identified biomarkers may not be reliable across different patient subgroups.

While the manuscript offers a promising approach to improving immunotherapy outcomes in NSCLC through spatial multi-omics, the substantial heterogeneity of the study cohort limits the strength and applicability of the conclusions. Addressing the dataset's variability and controlling for confounding factors are essential steps to validate the predictive models and ensure their reliability in clinical settings. By refining the study design/cohort, the authors can significantly enhance the robustness and clinical relevance of their findings.

Version 1:

Decision Letter:

IMPORTANT: Please note the reference number: NG-A66424R-Z Rimm. This number must be quoted whenever you communicate with us regarding this paper.

10th Apr 2025

Dear Dr Rimm,

Thank you for asking us to reconsider our decision on your manuscript "Enhancing Immunotherapy Outcomes: Spatial Multi-Omics Models for Non-Small Cell Lung Cancer". I have now discussed your appeal with my colleagues, and we think that you have some valid points. We therefore invite you to revise your manuscript along the lines that you propose.

When preparing a revision, please ensure that it fully complies with our editorial requirements for format and style; details can be found in the Guide to Authors on our website (<http://www.nature.com/ng/>).

Please be sure that your manuscript is accompanied by a separate letter detailing the changes you have made and your response to the points raised. At this stage we will need you to upload:

1) a copy of the manuscript in MS Word .docx format.

2) The Editorial Policy Checklist:

<https://www.nature.com/documents/nr-editorial-policy-checklist.pdf>

3) The Reporting Summary:

(Here you can read about the role of the Reporting Summary in reproducible science:

<https://www.nature.com/news/announcement-towards-greater-reproducibility-for-life-sciences-research-in-nature-1.22062>)

Please use the link below to be taken directly to the site and view and revise your manuscript:

Link Redacted

With kind wishes,

Safia Danovi, PhD

Senior Editor, Nature Genetics

ORCID: 0009-0007-7822-5479

Version 2:

Decision Letter:

21st May 2025

Dear Dr Rimm,

Your Article, "Enhancing Immunotherapy Outcomes: Spatial Multi-Omics Models for Non-Small Cell Lung Cancer" has now been seen by 2 referees. You will see from their comments below that while they find your work of interest, some important points are raised. We are interested in the possibility of publishing your study in Nature Genetics, but would like to consider your response to these concerns in the form of a revised manuscript before we make a final decision on publication.

To clarify, we are not requesting any further analyses. Rather, the aim here is to revise the narrative in a way that incorporates some of limitations voiced by Reviewer #2. In doing so, we are not suggesting that you unnecessarily undermine the importance of your findings, but encourage you to offer your reader a fair assessment of the 'clinical readiness' of the work. Our intention would be to assess the revisions in house - we'd only return to reviewers if absolutely necessary.

We therefore invite you to revise your manuscript taking into account all reviewer and editor comments. Please highlight all changes in the manuscript text file. At this stage we will need you to upload a copy of the manuscript in MS Word .docx or similar editable format.

*2) If you have not done so already please begin to revise your manuscript so that it conforms to our Article format instructions, available

[here](http://www.nature.com/ng/authors/article_types/index.html).

*3) Include a revised version of any required Reporting Summary: <https://www.nature.com/documents/nr-reporting-summary.pdf>

EXTENDED DATA FIGURES

Link Redacted

We hope to receive your revised manuscript within four to eight weeks. If you cannot send it within this time, please let us know.

Sincerely,

Safia Danovi, PhD
Senior Editor, Nature Genetics
ORCID: 0009-0007-7822-5479

Reviewers' Comments:

Reviewer #1 (Remarks to the Author):

Overall, the revised manuscript has addressed most of the prior critiques and is significantly improved. The authors should be commended for adding a second validation cohort with similar findings.

Several minor remaining comments include:

1. The abstract still references only two cohorts in the body. It would be helpful to the reader to now mention three cohorts.
2. In my prior comment #5, I questioned use of 2-year PFS. The authors' response is a bit confusing, as it focuses on recurrence rates and cites IMpower 010 and ADAURA as justification – both adjuvant studies. However, my understanding from the rest of the manuscript is that even though the baseline tissue samples are from resection specimens, the clinical outcomes are being measured in patients with "advanced/metastatic" disease treated in the first-line setting. In that context, recurrence rates (and studies of adjuvant therapy are irrelevant). Nonetheless, with the inclusion of the third-cohort showing similar findings, I think this is reasonable.
3. Likewise, the response to my prior comment #4 (supplemental Table 1) is unclear. If 50% of patients received "second-line therapy," then the number of patients in the "second-line or beyond" group should be at least 50% since "second-line or beyond" INCLUDES second-line. Based upon the numbers, I anticipate that this is 61%.

Reviewer #2 (Remarks to the Author):

The authors should be commended for their efforts to validate their findings in independent cohorts and for applying multivariable analysis to adjust for potential confounders. Despite the inherent challenges of cohort heterogeneity and limited sample sizes, they made a thoughtful attempt to account for key clinical variables, including age, sex, disease stage, histology, and treatment line. However, since the analysis seeks to identify prognostic factors associated with survival, the interpretability and robustness of the findings are still compromised by substantial heterogeneity across cohorts. Given these limitations, the claim that multivariable analysis confirms the independent prognostic value of the resistance and response signatures in the UQ and Greek cohorts is not well supported. UQ and Greek cohorts exhibit substantial imbalance across key clinical covariates. In the Greek cohort, for example, over 92% of patients were stage IV, and 86% received first-line nivolumab, creating limited variance in the very covariates the authors claim to adjust for. Similarly, the UQ cohort has a small sample size with minimal representation of histologic subtypes. These conditions severely restrict the ability of a multivariable model to adequately control for confounding, particularly when covariate overlap is minimal, and subgroup sizes are insufficient for reliable estimation. Moreover, the risk of overfitting and model instability is high. The lack of adjustment for potentially critical unmeasured variables, such as PD-L1 expression, TMB, performance status, and site-specific treatment practices further limits the interpretability of the findings. While the authors correctly emphasize the value of multi-omics profiling in advancing precision oncology, their conclusion that the model robustly defines resistance or predicts response in these cohorts overstates the capacity of the data. The small, heterogeneous, and imbalanced nature of the cohorts precludes strong claims of generalizability or independent prognostic significance. That said, the technical execution of the multi-omics assays and the analysis pipeline are exemplary. The analytical framework itself is robust and scientifically sound, and when applied to larger, better-balanced cohorts, is likely to yield highly informative and clinically actionable results.

Version 3:

Decision Letter:

Our ref: NG-A66424R2

2nd Jun 2025

Dear Dr Rimm,

Thank you for submitting your revised manuscript "Enhancing Immunotherapy Outcomes: Spatial Multi-Omics Models for Non-Small Cell Lung Cancer" (NG-A66424R2). It has now been seen by the original referees and their comments are below. The reviewers find that the paper has improved in revision, and therefore we'll be happy in principle to publish it in Nature Genetics, pending minor revisions to comply with our editorial and formatting guidelines.

Sincerely,

Safia Danovi, PhD
Senior Editor, Nature Genetics
ORCID: 0009-0007-7822-5479

Referee expertise: lung cancer, immunotherapy, spatial -omics.

Reviewers' Comments:

Reviewer #1 (Remarks to the Author):

In this manuscript, Aung, Monkman, Warrell and colleagues used a multi-omics approach leveraging spatial proteomics (CODEX) and transcriptomics (DSP-GeoMx-WTA) to profile the tumor immune microenvironment in two independent, pre-PD-1 non-small cell lung cancer datasets. Spatial proteomic analysis identified a resistance cell type model, characterized by proliferating tumor cells, granulocytes, and vessels. By contrast, a response cell type model was characterized by higher levels of M1 and M2 macrophages, as well as CD4 T cells. GeoMx was then used to derive gene signatures from these cell type models (e.g., resistance model – EMT and cell migration genes; response model – immunomodulatory genes).

Overall, this is a well-written and performed study. I had the following comments/questions:

Thank you for the comment.

1. The introduction emphasizes the importance of focusing on 1st line cohorts to minimize confounding from previous treatment regimens; however, the Queensland ICI cohort appears to include both 1st and 2nd line patients.

Response: Thank you for pointing this out. We acknowledge the importance of focusing on 1st line cohorts to minimize confounding from prior treatments. To further assess the generalizability of our identified signatures, we conducted additional spatial transcriptomic profiling on a newly analyzed cohort of 79 patients (Greece cohort), including 61 patients who received first-line immunotherapy exclusively. Both the Yale (training) and Greek (validation) cohorts consist exclusively of 1st line patients, and all signatures were trained on the Yale cohort. The Queensland ICI cohort, however, includes both 1st and 2nd line patients, with a small subset labeled as "second-line or beyond." Despite this, our signatures validated in both the UQ and 1st-line-only Greek cohorts. Furthermore, in our updated multivariable analysis, we show that the signatures retain significant predictive power when adjusting for line of treatment in the UQ cohort (see updated Figure 6, Supplementary Figure 16, and Supplementary Tables 4, 5, 6 and 7).

2. The Queensland cohort also consists of surgically resected specimens from patients who subsequently recurred. What steps were taken to ensure that these were recurrences versus new primary tumors (and thus that the specimens used in this analysis are representative of the pre-PD-1 tumors?)

Response: " Thank you for your question. The Queensland cohort consists of surgically resected specimens from NSCLC patients who later received ICIs in the advanced setting. All patients in this cohort were treated with ICI as second-line or later therapy."

3. While the manuscript and supplemental Table 1 references 113 patients in the Yale cohort, the actual number of patients with adequate tissue who met the eligibility criteria for inclusion was only 37-41. The second cohort was only 31 to 35 patients. Overall, these are relatively small sample sizes. It would also be advisable to provide the clinical characteristics of only those patients who were actually included in Table S1.

Response: The clinical characteristics of the patients analyzed in each cohort was now included in the Supplementary Table S2.

4. Supplemental Table 1 – In the Queensland group, 50% were treated in the 2nd line, but the table lists only 11.9% treated in "2nd-line or beyond." Presumably this should be "beyond 2nd-line"?

Response: The term "second-line or beyond" is correct because we were not certain about the exact treatment line beyond the second. This terminology accounts for patients who may have received second-line treatment as well as those who progressed to third-line or further therapies. It reflects the uncertainty in treatment sequencing while ensuring accurate representation of the data.

5. One major question is why the authors used 2-year PFS as their primary endpoint as opposed to objective response rate? Figure 2E-H treats this as a binary endpoint but obviously some patients will have much more durable disease control than others. Likewise, at various points in the manuscript, the authors use the

nomenclature of “responders” versus “non-responders” but it appears that they are referring to those who are progression-free versus those who are not progression-free.

Response: The reviewer correctly notes that objective response would be better than 2-year PFS. However objective response in a retrospectively collected cohort is a challenge to collect from patient records. To attempt better objectivity, we used PFS at 2 years. PFS-2 years provides a meaningful and clinically relevant measure of durable treatment benefit while ensuring reliable analysis. In lung cancer, particularly non-small cell lung cancer (NSCLC), recurrence rates are highest within the first 2–3 years after curative-intent therapy. Most recurrences in the adjuvant setting occur within this timeframe, making the 2-year mark a crucial point for evaluating treatment efficacy. Additionally, many landmark clinical trials in adjuvant lung cancer therapy, such as IMpower010 and ADAURA, have adopted 2-year PFS as a primary or secondary endpoint, underscoring its clinical significance. By using PFS-2 years, we align with established standards while ensuring a robust and timely assessment of therapeutic impact. We have corrected responders and non-responders to **progressors and non-progressors** in the revised manuscript as well as in **Fig. 4 and Supplementary Fig. 9**.

6. Proliferating tumor doesn't appear in Figure 2B. How was this then defined? Ki67? Other features?

Response: Proliferating tumor cells were defined using PanCK⁺ Ki67⁺ expressing cells. Please see the methods section “**Data integration, clustering, and phenotyping workflow**”, line number – 230.

7. One of the core features of the resistance cell type model is proliferating tumor cells. This is somewhat intuitive in that tumors with more highly proliferative tumor cells would presumably do worse. A central question is whether this is simply a prognostic factor and unrelated to IO outcomes? I would add that the KM curves in Figure 3F, while significant, are underwhelming, and show the curves crossing.

Response: Thank you for your comment. While proliferating tumor cells are likely a prognostic factor, their enrichment in the resistance cell type model suggests a potential role in immune evasion or IO resistance. To better assess whether proliferation is independently predictive of IO outcomes rather than just prognostic, we have performed additional analyses, including multivariate models adjusting for key confounders (see **Supplementary Table S3, S4, S5 and S6**). Additionally, we further explored gene expression patterns of the same patients beyond protein expression and cell typing. Please see **Figure 6**.

Regarding the crossing of the KM curves in Figure 3F, this may reflect underlying heterogeneity in the cohort OR simply because of the small number of IO treated patients in the UQ validation cohort. To address this, we have expanded our analysis by including an additional cohort to further evaluate the model's predictive power. Please see **Revised Figure 6 and Supplementary Figure 16**. We have also provided more details in the revised manuscript.

8. The validation of the response cell type model is also underwhelming, as the KMs appear to cross. On a related point, can the authors provide context for using a 1-sided log-rank for the UQ cohort and 2-sided for the Yale cohort?

Response: In the Yale cohort (training), we used a 2-sided log-rank test as we did not have prior expectations regarding the direction of the effects. However, for the UQ cohort and the Greek cohort (validation cohorts), we used a 1-sided log-rank test based on a predefined hypothesis of a directional effect (be it resistance or response) in line with the findings from the training cohort. Regarding the observed crossing of the Kaplan-Meier curves, this may reflect the inherent complexity of tumor-immune interactions and the heterogeneity within patient populations. We acknowledge this limitation and have clarified our statistical rationale in the **figure captions as well as in the revised manuscript (line number: 631), specifying the use of 2-sided versus 1-sided log-rank tests**.

9. Page 26, lines 4654-466: “The lack of statistical significance in OS further emphasizes the specificity of the response signature for predicting disease control and progression.” This is a rather optimistic view. An alternative explanation is that the model has modest predictive abilities that don't carry over to OS.

Response: Unlike progression-free survival (PFS), which is directly influenced by treatment status, OS is impacted by multiple additional factors, including patient age, comorbidities, and non-cancer-related complications. To further evaluate the model's predictive power, we tested additional cohorts for both PFS and OS. While we observed a consistent trend between OS and PFS, the lack of statistical significance in OS is likely due to the influence of non-cancer-related mortality, making it inherently more variable than PFS. Please see **Supplementary Figs. S14, S15 and S16**. Nonetheless, the observed trend suggests that these signatures are associated with patient outcomes.

10. M2 and PD-L1+ appear to be in opposite directions of what has been seen clinically.

Response: Thank you for your comment. Our results only conflict partially with previous clinical findings. Rather, the prognostic significance of PD-L1 protein expression on tumor cells appears to be inconsistent across two cohorts. In our analysis, while PD-L1 expression on tumor cells was a statistically significant prognostic factor in one cohort (**Yale Cohort, Figure 4I**), this significance diminished when analyzing the combined cohort (**Supplementary Fig. S9C**). In contrast, PD-L1 expression on macrophages remained a significant factor associated with response to immunotherapy (**see Figure 4J and Supplementary Fig. S9D**). To further support this, we have included t-SNE plots in **Figure 4H** to visualize these cell populations and their distribution in our dataset. Additionally, we have revised our analysis and **updated Figure 4 and Supplementary Figure 9** to provide further clarity. These results are discussed in detail in the revised manuscript.

11. Page 29, lines 518, “These findings suggest that the presence of PD-L1 on M1 and M2 macrophages, rather than on tumor cells, could be indicative of a positive response to treatment.” While the discussion section on the benefits of CPS versus TPS are certainly reasonable, I would caution the authors from over-interpreting their data given the very small sample size – especially since they didn’t show that tumor cell PD-L1 positive correlated with outcomes (but it has in many large phase III clinical trials).

Response: Thank you for your comment. Please refer to our response above regarding the interpretation of PD-L1 expression on tumor cells and macrophages. We acknowledge the limitations of our sample size and have revised our analyses to directly compare PD-L1 expression on tumor cells and macrophages between progressors and non-progressors. The updated results can be found in revised **Figure 4 and Supplementary Figure 9**. These findings are now discussed in greater detail in the revised manuscript to provide a more balanced interpretation in the context of existing large phase III clinical trials.

12. Figure 5c – There is a fair bit of overlap in M2 fractions between those with progression and those without progression.

Response: Our model was constructed using protein expression data, and Figure 5C serves as a validation of the findings from the protein-based model. For this validation, we used RNA transcriptomics to infer cell types through computational deconvolution, which inherently introduces some variability. While there is overlap in M2 macrophage fractions between progressors and non-progressors, the observed differences remain statistically significant. We have clarified this distinction in the revised manuscript to ensure the context and limitations of the analysis are clearly communicated.

13. In the limitations section, it’s the first time we see it mentioned that some patients received chemo-immunotherapy but it’s unclear how many patients? Supplemental Table 1 doesn’t list chemo-immunotherapy.

Response: We have revised the limitations section to provide clarity on the inclusion of patients who received prior chemotherapy and chemo-immunotherapy. Additionally, we have updated **Supplemental Table 1 and Table 2** to specify the number of patients treated with chemo-immunotherapy. To account for its potential influence on treatment outcomes, we also conducted multivariable analyses adjusting for prior chemotherapy as a confounding factor. These revisions ensure a more comprehensive assessment of the impact of prior chemotherapy on our findings.

Reviewer #2 (Remarks to the Author):

Enhancing Immunotherapy Outcomes: Spatial Multi-Omics Models for Non-Small Cell Lung Cancer by Aung and Monkman, et al. presents an innovative approach to predicting patient responses to immunotherapy in non-small cell lung cancer. By integrating spatial proteomic and transcriptomic profiling, the study aims to identify specific cell types and gene signatures within the tumor immune microenvironment associated with treatment resistance and response. The study’s strengths include its multi-omics approach and the application of machine learning, both of which have the potential to enhance precision oncology.

Response: Thank you for the positive feedback. We appreciate your recognition of the strengths in our multi-omics approach and the application of machine learning to enhance precision oncology.

The major significant limitation of this study is the heterogeneity of the dataset. The cohort includes patients with varying treatment regimens, disease stages, and other clinical variables such as age, gender, ethnicity,

smoking history, performance status, comorbidities, tumor stage, and prior treatments. Additionally, differences in immunotherapy types, dosages, and concomitant treatments further contribute to the variability. This introduces numerous confounding variables that complicate the interpretation of the results. Without adequate control for these factors—such as through multivariable regression analysis or stratification—the associations between immune microenvironment features and clinical outcomes may be confounded by unaccounted influences. Consequently, the predictive value of the identified biomarkers may not be reliable across different patient subgroups.

Response: We acknowledge the heterogeneity of our dataset, including variations in clinical characteristics, treatment regimens, and other factors that contribute to the complexity of the analysis. However, this diversity reflects real-world clinical scenarios, thereby enhancing the practical relevance of our findings. Additionally, to further assess the generalizability of our identified signatures, we performed additional spatial transcriptomic profiling on a new cohort of 79 patients (Greek cohort), including 61 patients who received first-line treatment only.

To address potential confounding factors, we conducted comprehensive multivariable analyses, adjusting for all relevant clinical variables, as detailed in **Supplementary Tables S4, S5, S6 and S7. Please see the results in line: 626 and line: 653.** These analyses, including Cox regression and chi-square tests, confirm that the identified signatures remain statistically significant after controlling for multiple covariates.

Furthermore, we have expanded our discussion on the limitations regarding the predictive value of these biomarkers across different patient subgroups and incorporated the additional cohort into all validation tests.

The updated analyses and findings are presented in revised **Figure 4 and Figure 6**, as well as **Supplementary Figs. S9, S14, S15 and S16.**

While the manuscript offers a promising approach to improving immunotherapy outcomes in NSCLC through spatial multi-omics, the substantial heterogeneity of the study cohort limits the strength and applicability of the conclusions. Addressing the dataset's variability and controlling for confounding factors are essential steps to validate the predictive models and ensure their reliability in clinical settings. By refining the study design/cohort, the authors can significantly enhance the robustness and clinical relevance of their findings.

Response: We appreciate the recognition of our approach in leveraging spatial multi-omics to enhance immunotherapy outcomes in NSCLC. Regarding cohort heterogeneity and the control of confounding factors, we have addressed these concerns through comprehensive multivariable analyses (see response above). To further strengthen the robustness and clinical relevance of our findings, we incorporated the additional cohort (N=79) and analyzed the patients only treated with **1st line immunotherapy (N=61)** for validation. These efforts are discussed in the revised manuscript to provide a more comprehensive assessment of the predictive models.

Point by point response to the reviewers

Reviewer #1

1. The abstract still references only two cohorts in the body. It would be helpful to the reader to now mention three cohorts.

Response:

We thank the reviewer for pointing this out. We have revised the abstract to indicate that three cohorts were included in our study. The updated sentence now reads:

Abstract: " We studied three independent cohorts of advanced NSCLC tissue samples, treated with PD–1–based immunotherapies."

2. In my prior comment #5, I questioned use of 2-year PFS. The authors' response is a bit confusing, as it focuses on recurrence rates and cites IMpower 010 and ADAURA as justification – both adjuvant studies. However, my understanding from the rest of the manuscript is that even though the baseline tissue samples are from resection specimens, the clinical outcomes are being measured in patients with "advanced/metastatic" disease treated in the first-line setting. In that context, recurrence rates (and studies of adjuvant therapy are irrelevant).

Response:

We sincerely appreciate the reviewer's thoughtful comment. We would like to clarify that while IMpower010 and ADAURA were not directly cited in the manuscript, we recognize that our original explanation may have inadvertently suggested an adjuvant context by referencing recurrence rates. To clarify, all patients in our study were treated with immunotherapy in the advanced/metastatic setting. The choice of a 2-year PFS endpoint was informed by its clinical relevance in this context. Specifically, pivotal trials such as KEYNOTE-189 and KEYNOTE-407 limited immunotherapy duration to two years, and recent real-world data have demonstrated no significant survival benefit with treatment beyond that time point. Therefore, we believe that 2-year PFS is a meaningful and appropriate endpoint for evaluating long-term benefit in this setting. We have revised the Methods section of the manuscript to reflect this clarification.

Original sentence:

"The clinical endpoint was set to PFS at 2 years, which was assessed in both the training and validation cohorts, minimizing the effects of secondary malignancies and subsequent lines of treatment."

Revised version:

Methods: "The clinical endpoint was set to 2-year PFS, consistent with prior real-world and clinical trial data supporting this timeframe as a clinically meaningful surrogate for durable benefit in advanced/metastatic NSCLC. Notably, major immunotherapy trials that led to regulatory approval, such as KEYNOTE-189 and KEYNOTE-407, implemented a 2-year cap on treatment duration (Gandhi et al., 2018; Paz-Ares et al., 2021). In real-world clinical practice, treatment duration beyond two years was common due to the absence of data defining optimal therapy length. However, recent retrospective evidence from over 1,000 patients demonstrated no statistically significant overall survival benefit for those treated beyond two years compared to those who stopped at that point (Sun et al., 2023). Furthermore, approximately 20% of patients in that study discontinued treatment at two years despite the absence of progression. Taken together with the toxicity and financial burdens of prolonged therapy, these findings support a shift toward 2-year treatment as a clinical standard. Therefore, adopting 2-year PFS as an endpoint enhances the real-world relevance and interpretability of our results within the context of advanced/metastatic disease treated in the first-line setting."

3. Likewise, the response to my prior comment #4 (supplemental Table 1) is unclear. If 50% of patients received "second-line therapy," then the number of patients in the "second-line or beyond" group should be at least 50% since "second-line or beyond" INCLUDES second-line. Based upon the numbers, I anticipate that this is 61%.

Response:

We sincerely thank the reviewer for this observation. Upon further review, we observed that we had not fully clarified how the "second-line or beyond" category was derived in our original table. As the reviewer rightly noted, this should include all patients treated with immunotherapy beyond the first-line setting.

We have now recalculated this by including all patients who received second-line, third-line, fourth-line, adjuvant, consolidation, neoadjuvant, or unclassified/unknown lines of therapy. The updated values for “second-line or beyond” immunotherapy are:

1. 53.1% for the Yale cohort
2. 78.6% for the UQ cohort
3. 13.6% for the Greek cohort

These values have now been updated in Supplementary Table 1. We appreciate the reviewer’s careful attention to this detail.

Reviewer #2

The authors should be commended for their efforts to validate their findings in independent cohorts and for applying multivariable analysis to adjust for potential confounders. Despite the inherent challenges of cohort heterogeneity and limited sample sizes, they made a thoughtful attempt to account for key clinical variables, including age, sex, disease stage, histology, and treatment line. However, since the analysis seeks to identify prognostic factors associated with survival, the interpretability and robustness of the findings are still compromised by substantial heterogeneity across cohorts.

Response:

We thank the reviewer for their thoughtful and detailed feedback. We agree that the heterogeneity and limited sample sizes of the validation cohorts present important limitations for interpreting the generalizability and independent prognostic significance of our findings. In response, we have revised both the Methods and Discussion sections of the manuscript to directly address these concerns, and we outline how these respond to the specific points below.

Given these limitations, the claim that multivariable analysis confirms the independent prognostic value of the resistance and response signatures in the UQ and Greek cohorts is not well supported. UQ and Greek cohorts exhibit substantial imbalance across key clinical covariates. In the Greek cohort, for example, over 92% of patients were stage IV, and 86% received first-line nivolumab, creating limited variance in the very covariates the authors claim to adjust for. Similarly, the UQ cohort has a small sample size with minimal representation of histologic subtypes.

Response:

We recognize that the balance of covariates in the UQ and Greek cohorts is different from the Yale cohort. As the reviewer states, over 92% of patients were stage IV in the Greek cohort; however, while 86% of the Greek patients received first-line immunotherapy, not all of these were treated with Nivolumab, and a substantial proportion were treated with Pembrolizumab, as in the Yale cohort. While the restricted composition of the validation cohorts does not fully establish generalization in a highly heterogeneous setting, we were motivated in our selection of these cohorts by the intention to demonstrate the applicability of the signatures in first line immunotherapy treated patients, the proportion of which are enriched in both UQ and Greek cohorts. We indicate below the sections of the manuscript which have been modified to emphasize the issues above.

Revised text:

Methods: “These patients were included to further validate the robustness and generalizability of the findings derived from the Yale training cohort, although we note that the balance of covariates in these cohorts differs from the Yale cohort, particularly in the increased proportion of Stage IV patients in both Greece and UQ cohorts.”

Results: “Multivariable analysis, adjusting for age, sex, disease stage, prior chemotherapy, type of immunotherapy, line of immunotherapy, histology, and smoking status, confirmed that the resistant signature remained statistically significantly associated with worse outcomes in both the UQ and Greek cohorts (Supplementary Tables S4 and S5). However, we note that the Greek cohort was composed of over 92% stage IV patients and 86% receiving first-line immunotherapy, creating limited variance across this covariate. Similarly, the UQ cohort had a small sample size and underrepresentation of some histologic subtypes. These constraints may limit the capacity of multivariable models to effectively control for confounding, as discussed below”

Discussion: “While our findings provide strong support for the potential of multi-omics profiling to predict outcomes in immunotherapy treated NSCLC, we acknowledge a number of limitations. Notably, differences in

the proportion of disease stages, treatment regimens and line of immunotherapy in validation cohorts, constrain the interpretability of multivariable results. The Greek cohort consisted predominantly of stage IV patients with first line immunotherapy treatment, and the UQ cohort included a limited representation of some subtypes. We intentionally selected the Greek cohort for its enrichment in first line immunotherapy treated patients, since we intended to demonstrate the applicability of our signatures in this setting. However, we acknowledge that the broad generalizability of our signatures in highly heterogeneous cohorts requires further testing.”

These conditions severely restrict the ability of a multivariable model to adequately control for confounding, particularly when covariate overlap is minimal, and subgroup sizes are insufficient for reliable estimation. Moreover, the risk of overfitting and model instability is high. The lack of adjustment for potentially critical unmeasured variables, such as PD-L1 expression, TMB, performance status, and site-specific treatment practices further limits the interpretability of the findings.

Response:

We acknowledge that the reduced number of patients in certain covariate subgroups in the validation cohorts may limit the accuracy of the effect sizes estimated for certain variables in the multivariable analysis (for instance, Durvalumab in the UQ analysis). Likewise, the fact that variables such as PD-L1 expression, tumor mutation burden, and performance status were not consistently available means that these could be potential confounding effects. However, the limited variation in line of immunotherapy in the Greece cohort is only limiting in the context of validating our signatures in a highly heterogeneous setting; as noted above, we were interested in testing the prognostic potential of our signatures in a first line immunotherapy setting, which motivated our use of the Greece cohort for validation as noted above. Also, we note that, despite the limitations noted above, our resistance and response signatures achieve statistical significance in the multivariable analyses of both UQ and Greece cohorts while controlling for all available covariates, suggesting that they generalize in these (potentially limited) contexts. We indicate below the sections of the manuscript which have been modified to emphasize these issues

Revised text:

Discussion: “Further, we acknowledge that, due to the reduced number of patients in certain covariate subgroups in the validation cohorts, the multivariable analysis may not be able to quantify the effects of all covariates with high accuracy, so an accurate determination of the effect sizes will benefit from testing on larger cohorts. Additionally, variables such as PD-L1 expression, tumor mutation burden, and performance status were not consistently available and could not be included in our models. ”

Discussion: “Due to the small number of patients who exclusively received first-line immunotherapy in the UQ validation cohort, we included patients treated with first–line, second–line, and second–line or beyond immunotherapy, introducing some heterogeneity to assess the statistical association of our signatures in this context.”

While the authors correctly emphasize the value of multi-omics profiling in advancing precision oncology, their conclusion that the model robustly defines resistance or predicts response in these cohorts overstates the capacity of the data. The small, heterogeneous, and imbalanced nature of the cohorts precludes strong claims of generalizability or independent prognostic significance. That said, the technical execution of the multi-omics assays and the analysis pipeline are exemplary. The analytical framework itself is robust and scientifically sound, and when applied to larger, better-balanced cohorts, is likely to yield highly informative and clinically actionable results.

Response:

We appreciate the reviewer’s recognition of the technical rigor of our analytic framework and believe these revisions enhance the transparency and integrity of the study. We emphasize in our revised text that our conclusions regarding generalizability and independent prognostic value are to be understood with appropriate caution, and we also highlight the need for prospective validation in more heterogeneous cohorts to fully establish clinical utility.

Revised text:

Discussion: “The ability of our models to achieve statistical significance within this heterogeneous treatment setting adds credibility to their broad potential clinical utility, although as noted above, the generalizability of the signatures as prognostic models requires further validation in more heterogeneous datasets.”